# An optimal control perspective
# on diffusion-based generative modeling

**Julius Berner**$^{\star\dagger}$     *jberner@caltech.edu*
*Caltech*

**Lorenz Richter**$^{\star}$     *richter@zib.de*
*Zuse Institute Berlin*
*dida Datenschmiede GmbH*

**Karen Ullrich**     *karenu@meta.com*
*Meta AI*

**Reviewed on OpenReview:** *https://openreview.net/forum?id=oYIjw37pTP*

## Abstract

We establish a connection between stochastic optimal control and generative models based on stochastic differential equations (SDEs), such as recently developed diffusion probabilistic models. In particular, we derive a Hamilton–Jacobi–Bellman equation that governs the evolution of the log-densities of the underlying SDE marginals. This perspective allows to transfer methods from optimal control theory to generative modeling. First, we show that the evidence lower bound is a direct consequence of the well-known verification theorem from control theory. Further, we can formulate diffusion-based generative modeling as a minimization of the Kullback–Leibler divergence between suitable measures in path space. Finally, we develop a novel diffusion-based method for sampling from unnormalized densities – a problem frequently occurring in statistics and computational sciences. We demonstrate that our time-reversed diffusion sampler (DIS) can outperform other diffusion-based sampling approaches on multiple numerical examples.

## 1 Introduction

*Diffusion (probabilistic) models* (DPMs) have established themselves as state-of-the-art in generative modeling and likelihood estimation of high-dimensional image data (Ho et al., 2020; Kingma et al., 2021; Nichol & Dhariwal, 2021). In order to better understand their inner mechanisms, multiple connections to adjacent fields have already been proposed: For instance, in a discrete-time setting, one can interpret DPMs as types of *variational autoencoders* (VAEs) for which the *evidence lower bound* (ELBO) corresponds to a multi-scale denoising score matching objective (Ho et al., 2020). In continuous time, DPMs can be interpreted in terms of stochastic differential equations (SDEs) (Song et al., 2021; Huang et al., 2021; Vahdat et al., 2021) or as infinitely deep VAEs (Kingma et al., 2021). Both interpretations allow for the derivation of a continuous-time ELBO, and encapsulate various methods, such as *denoising score matching with Langevin dynamics* (SMLD), *denoising diffusion probabilistic models* (DDPM), and continuous-time *normalizing flows* as special cases (Song et al., 2021; Huang et al., 2021).

In this work, we suggest another perspective. We show that the SDE framework naturally connects diffusion models to partial differential equations (PDEs) typically appearing in *stochastic optimal control* and *reinforcement learning*. The underlying insight is that the *Hopf–Cole transformation* serves as a means of

---

$^{\star}$Equal contribution (the author order was determined by `numpy.random.rand(1)`).

$^{\dagger}$Work done during an internship at Meta AI.

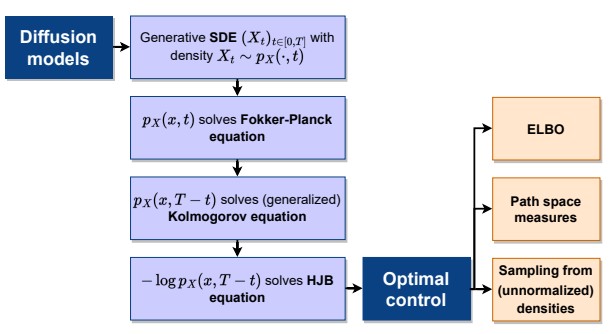

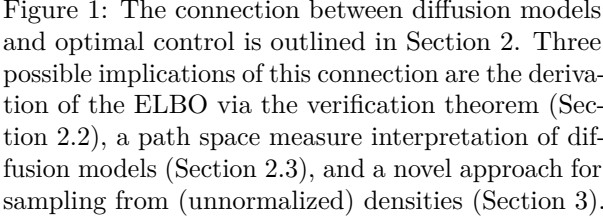

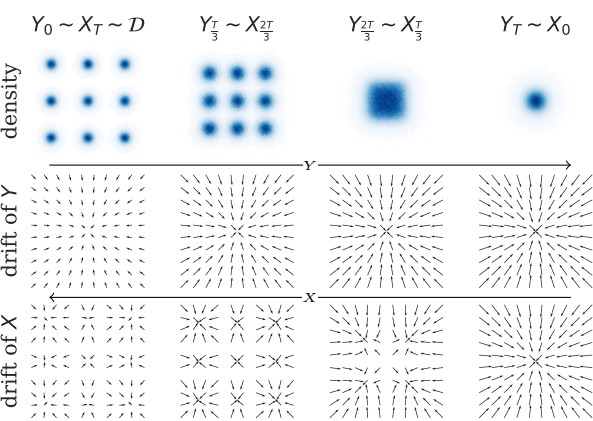

Figure 1: The connection between diffusion models and optimal control is outlined in Section 2. Three possible implications of this connection are the derivation of the ELBO via the verification theorem (Section 2.2), a path space measure interpretation of diffusion models (Section 2.3), and a novel approach for sampling from (unnormalized) densities (Section 3).

Figure 2: For a stochastic process $Y$ and its time-reversed process $X$, we show the density $p_Y(\cdot, t) = p_X(\cdot, T-t)$ (top), the drift $f(\cdot, t)$ of $Y$ (middle), and the drift $\mu(\cdot, t)$ of $X$ (bottom) for $t \in \left[0, \frac{T}{3}, \frac{2T}{3}, T\right]$, see (2) and (3).

showing that the time-reversed log-density of the diffusion process satisfies a *Hamilton–Jacobi–Bellman* (HJB) equation (Section 2.1). This relation allows us to connect diffusion models to a control problem that minimizes specific control costs with respect to a given controlled dynamics. We show that this readily yields the ELBO of the generative model (Section 2.2), which can subsequently be interpreted in terms of Kullback-Leibler (KL) divergences between measures on the path space of continuous-time stochastic processes (Section 2.3).

While one of our contributions lies in the formal connection between stochastic optimal control and diffusion models, as described in Section 2.4, we moreover demonstrate the practical relevance of our analysis by transferring methods from control theory to generative modeling. More specifically, in Section 3, we design a novel algorithm for sampling from (unnormalized) densities – a problem which frequently occurs in Bayesian statistics, computational physics, chemistry, and biology (Liu & Liu, 2001; Stoltz et al., 2010). As opposed to related approaches (Dai Pra, 1991; Richter, 2021; Zhang & Chen, 2022a), our method allows for more flexibility in choosing the initial distribution and reference SDE, offering the possibility to incorporate specific prior knowledge. Finally, in Section 4, we show that our sampling strategy can significantly outperform related approaches across a number of relevant numerical examples.

In summary, our work establishes new connections between generative modeling, sampling, and optimal control, which brings the following theoretical insights and practical algorithms (see Figure 1 for an overview):

- **PDE and optimal control perspectives:** We identify the HJB equation as the governing PDE of the score function, which rigorously establishes the connection of generative modeling to optimal control (Section 2.1). The HJB equation can further be used for theoretic analyses as well as for practical algorithms for the numerical approximation of the score, e.g., based on neural PDE solvers.

- **ELBO:** Using the HJB equation, we show that the objectives in diffusion-based generative modeling can be solely derived from the principles of control theory (Section 2.2).

- **Path space perspective:** This perspective on diffusion models offers an intuitive representation of the objective (and the variational gap) in terms of KL divergences between measures on path space (Section 2.3). Crucially, it also allows to consider alternative divergences, such as, e.g., the log-variance divergence, leading to improved loss functions and algorithms.

- **Sampling:** Our control perspective leads to a novel, diffusion-based method to sample from unnormalized densities (Section 3). This method can already outperform previous diffusion-based state-of-the-art samplers. Moreover, the connection to diffusion models allows to transfer noise schedules, integrators, and further popular techniques from generative modeling to sampling.

## 1.1 Related work

**Diffusion models:** Starting from DPMs (Sohl-Dickstein et al., 2015), a number of works have contributed to the success of diffusion-based generative modeling, see, e.g., Ho et al. (2020); Kingma et al. (2021); Nichol & Dhariwal (2021); Vahdat et al. (2021); Song & Ermon (2020). We are building upon the SDE-based formulation developed by Song et al. (2021) which connects diffusion models to score matching (Hyvärinen & Dayan, 2005). The underlying idea of time-reversing a stochastic process dates back to work by Nelson (1967); Anderson (1982); Haussmann & Pardoux (1986); Föllmer (1988). Notably, Pavon (1989) connects the log-density of such a reverse-time SDE to an HJB equation. In this setting, we extend the results of Huang et al. (2021) on the (continuous-time) ELBO of diffusion models and provide further insights from the perspective of optimal control and path space measures.

For further previous work on optimal control in the context of generative modeling, we refer the reader to Tzen & Raginsky (2019); De Bortoli et al. (2021); Pavon (2022); Holdijk et al. (2022). Connections between optimal control and time-reversals of stochastic processes have also been analyzed by Maoutsa & Opper (2022); Maoutsa (2023). Finally, we refer to Richter & Berner (2024) for further generalizations based on the path space perspective and to Zhang & Katsoulakis (2023) for recent work on interpreting generative modeling via mean-field games.

**Sampling from (unnormalized) densities:** Monte Carlo (MC) techniques are arguably the most common methods to sample from unnormalized densities and compute normalizing constants. Specifically, variations of *Annealed Importance Sampling* (AIS) (Neal, 2001) and *Sequential Monte Carlo* (Del Moral et al., 2006; Doucet et al., 2009) (SMC) are often referred to as the "gold standard" in the literature. They apply a combination of Markov chain Monte Carlo (MCMC) methods and importance sampling to a sequence of distributions interpolating between a tractable initial distribution and the target density. Even though MCMC methods are guaranteed to converge to the target distribution under mild assumptions, the convergence speed might be too slow in many practical settings (Robert et al., 1999). Variational methods such as mean-field approximations (Wainwright et al., 2008) and normalizing flows provide an alternative[1] (Papamakarios et al., 2021). By fitting a parametric family of tractable distributions to the target density, the problem of density estimation is cast into an optimization problem. In Salimans et al. (2015) it is noted that stochastic Markov chains can be interpreted as the variational approximation in an extended state space, thus bridging the two techniques and allowing to apply them jointly.

In this context, diffusion models have been employed to approximate the extended target distribution needed in the importance sampling step of AIS methods (Doucet et al., 2022). Moreover, diffusion models have been trained on densities by simulating samples using importance sampling with the likelihood of the partially-trained model (computed via the probability flow ODE) as proposal distribution (Jing et al., 2022). In this work, we propose a novel variational method based on diffusion models, which we call *time-reversed diffusion sampler* (DIS). It is based on minimizing the reverse KL divergence between a controlled SDE and the reverse-time SDE. This is intimately connected to methods developed in the field of stochastic optimal control based on minimizing the reverse KL divergence to a reference process starting at a Dirac delta distribution (Dai Pra, 1991; Richter, 2021; Zhang & Chen, 2022a; Tzen & Raginsky, 2019; Vargas et al., 2023b). Finally, we mention that concurrently to our work, *Denoising Diffusion Samplers* (Vargas et al., 2023a) have been proposed, which can be viewed as a special case of our approach, see Section A.10.

**Schrödinger bridges:** Schrödinger bridge (SB) problems (Schrödinger, 1931) aim to find the minimizer of the KL divergence to a reference process, typically a Brownian motion, subject to the constraint of satisfying given marginal distributions at the initial and terminal time. Such problems include diffusion-based methods as a special case where the reference process is given by the uncontrolled inference process (De Bortoli et al., 2021; Chen et al., 2021; Koshizuka & Sato, 2022). Since in this case only the reverse-time process is controlled, its initial distribution must correspond to the terminal distribution of the inference process. In practice, this constraint is usually only fulfilled in an approximate sense, resulting in a prior loss, see Section 2.3. Note that the previously mentioned sampling methods in the field of optimal control rely on a Dirac delta as the

---

[1]Note that diffusion models give rise to a continuous-time normalizing flow, the so-called *probability flow ordinary differential equation (ODE)*, having the same marginals as the reverse-time diffusion process, see Song et al. (2021) and Section A.3.

initial distribution and solve the so-called *Schrödinger half-bridge problem*, constraining the KL minimization problem only to the terminal distribution (Dai Pra, 1991).

## 1.2 Notation

We denote the density of a random variable $Z$ by $p_Z$. For an $\mathbb{R}^d$-valued stochastic process $Z = (Z_t)_{t \in [0,T]}$ we define the function $p_Z$ by $p_Z(\cdot, t) \coloneqq p_{Z_t}$ for every $t \in [0, T]$. We further denote by $\mathbb{P}_Z$ the law of $Z$ on the space of continuous functions $C([0,T], \mathbb{R}^d)$. For a time-dependent function $f$, we denote by $\overleftarrow{f}$ the time-reversal given by $\overleftarrow{f}(t) \coloneqq f(T - t)$. Finally, we define the divergence of matrix-valued functions row-wise. More details on our notation can be found in Section A.1.

## 2 SDE-based generative modeling as an optimal control problem

Diffusion models can naturally be interpreted through the lens of continuous-time stochastic processes (Song et al., 2021). To this end, let us formalize our setting in the general context of SDE-based generative models. We define our model as the stochastic process $X = (X_s)_{s \in [0,T]}$ characterized by the SDE

$$\mathrm{d}X_s = \bar{\mu}(X_s, s)\,\mathrm{d}s + \bar{\sigma}(s)\,\mathrm{d}B_s \tag{1}$$

with suitable[2] drift and diffusion coefficients $\mu \colon \mathbb{R}^d \times [0, T] \to \mathbb{R}^d$ and $\sigma \colon [0, T] \to \mathbb{R}^{d \times d}$. Learning the model in (1) now corresponds to solving the following problem.

**Problem 2.1** (SDE-based generative modeling)**.** Learn an initial condition $X_0$ as well as coefficient functions $\mu$ and $\sigma$ such that the distribution of $X_T$ approximates a given data distribution $\mathcal{D}$.

While, in general, the initial condition $X_0$ as well as both the coefficient functions $\mu$ and $\sigma$ can be learned, typical applications often focus on learning only the drift $\mu$ (Ho et al., 2020; Kong et al., 2021; Corso et al., 2022). The following remark justifies that this is sufficient to represent an arbitrary distribution $\mathcal{D}$.

*Remark* 2.2 (Reverse-time SDE)**.** Naively, one can achieve $X_T \sim \mathcal{D}$ by setting

$$X_0 \sim Y_T \quad \text{and} \quad \mu \coloneqq \sigma\sigma^\top \nabla \log p_Y - f, \tag{2}$$

where $f \colon \mathbb{R}^d \times [0, T] \to \mathbb{R}^d$ and $Y$ is a solution to the SDE

$$\mathrm{d}Y_s = f(Y_s, s)\,\mathrm{d}s + \sigma(s)\,\mathrm{d}B_s, \quad Y_0 \sim \mathcal{D}. \tag{3}$$

This well-known result dates back to Nelson (1967); Anderson (1982); Haussmann & Pardoux (1986); Föllmer (1988). More generally, it states that $X$ can be interpreted as the time-reversal of $Y$, which we will denote by $\overleftarrow{Y}$, in the sense that $p_X = \overleftarrow{p}_Y$ almost everywhere, see Figure 2 and Section A.3. Even though the reverse-time SDE provides a valid solution to Problem 2.1, apparent practical challenges are to sample from the terminal distribution of the generative SDE, $X_0 \sim Y_T$, and to compute the so-called *score* $\nabla \log p_Y$. In most scenarios, we either only have access to samples from the true data distribution $\mathcal{D}$ or access to its density. If samples from $\mathcal{D}$ are available, diffusion models provide a working solution to solving Problem 2.1 (more details in Section 2.4). When, instead, having access to the density of $\mathcal{D}$, general time-reversed diffusions have, to the best of our knowledge, not yet been considered. In Section 3, we thus propose a novel strategy for this scenario.

As already apparent from the optimal drift $\mu$ in the previous remark, and noting again that $p_Y = \overleftarrow{p}_X$ in this case, the reverse-time log-density $\log \overleftarrow{p}_X$ of the process $X$ will play a prominent role in deriving a suitable objective for solving Problem 2.1. Hence, in the next section, we derive the HJB equation governing the evolution of $\log \overleftarrow{p}_X$, which then provides the bridge to the fields of optimal control and reinforcement learning.

---

[2]Motivated by Remark 2.2, we start with time-reversed drift and diffusion coefficients $\bar{\mu}$ and $\bar{\sigma}$. Further, we assume certain regularity on the coefficient functions of all appearing SDEs, see Section A.1

## 2.1 PDE perspective: HJB equation for log-density

We start with the well-known *Fokker-Planck equation*, which describes the evolution of the density of the solution $X$ to the SDE in (1) via the PDE

$$\partial_t p_X = \text{div} \left( \text{div} \left( \breve{D} p_X \right) - \bar{\mu} p_X \right), \tag{4}$$

where we set $D := \frac{1}{2} \sigma \sigma^\top$ for notational convenience. This implies that the time-reversed density $\breve{p}_X$ satisfies a (generalized) *Kolmogorov backward equation* given by

$$\partial_t \breve{p}_X = \text{div} \left( - \text{div} \left( D \breve{p}_X \right) + \mu \breve{p}_X \right) = - \text{Tr} \left( D \nabla^2 \breve{p}_X \right) + \mu \cdot \nabla \breve{p}_X + \text{div}(\mu) \breve{p}_X. \tag{5}$$

The second equality follows from the identities for divergences in Section A.2 and the fact that $\sigma$ does not depend on the spatial variable[3] $x$. Now we use the *Hopf–Cole transformation* to convert the linear PDE in (5) to an HJB equation which is prominent in control theory, see Pavon (1989); Fleming & Rishel (2012), and Section A.5.

**Lemma 2.3** (HJB equation for log-density)**.** *Let us define $V := - \log \breve{p}_X$. Then $V$ is a solution to the HJB equation*

$$\partial_t V = - \text{Tr} \left( D \nabla^2 V \right) + \mu \cdot \nabla V - \text{div}(\mu) + \frac{1}{2} \left\| \sigma^\top \nabla V \right\|^2 \tag{6}$$

*with terminal condition $V(\cdot, T) = - \log p_{X_0}$.*

For approaches to directly solve Kolmogorov backward or HJB equations via deep learning in rather high dimensions, we refer to, e.g., Richter (2021); Berner et al. (2020); Zhou et al. (2021); Nüsken & Richter (2023; 2021); Richter & Berner (2022), see also Sections A.4 and A.11. In the following, we leverage the HJB equation and tools from stochastic control theory to derive a suitable objective for Problem 2.1.

## 2.2 Optimal control perspective: ELBO derivation

We will derive the ELBO for our generative model (1) using the following fundamental result from control theory, which shows that the solution to an HJB equation, such as the one stated in Lemma 2.3, is related to an optimal control problem, see Dai Pra (1991); Pavon (1989); Nüsken & Richter (2021); Fleming & Soner (2006); Pham (2009) and Section A.7. For a general introduction to stochastic optimal control theory we refer to Section A.6.

**Theorem 2.4** (Verification theorem)**.** *Let $V$ be a solution to the HJB equation in Lemma 2.3. Further, let $\mathcal{U} \subset C^1(\mathbb{R}^d \times [0,T], \mathbb{R}^d)$ be a suitable set of admissible controls and for every control $u \in \mathcal{U}$ let $Y^u$ be the solution to the controlled SDE[4]*

$$\mathrm{d}Y_s^u = (\sigma u - \mu)(Y_s^u, s)\,\mathrm{d}s + \sigma(s)\,\mathrm{d}B_s. \tag{7}$$

*Then it holds almost surely that*

$$V(Y_0^u, 0) = \min_{u \in \mathcal{U}} \mathbb{E} \left[ \mathcal{R}_\mu^u(Y^u) - \log p_{X_0}(Y_T^u) \big| Y_0^u \right], \tag{8}$$

*where $- \log p_{X_0}(Y_T^u)$ constitutes the terminal costs, and the running costs are defined as*

$$\mathcal{R}_\mu^u(Y^u) := \int_0^T \left( \text{div}(\mu) + \frac{1}{2} \|u\|^2 \right)(Y_s^u, s)\,\mathrm{d}s. \tag{9}$$

*Moreover, the unique minimum is attained by $u^* := - \sigma^\top \nabla V$.*

---

[3]In Section A.3, we provide a proof for the reverse-time SDE in Remark 2.2 for general $\sigma$ depending on $x$ and $t$. However, for simplicity, we restrict ourselves to $\sigma$ only depending on the time variable $t$ in the following.

[4]As usually done, we assume that the initial condition $Y_0^u$ of a solution $Y^u$ to a controlled SDE does not depend on the control $u$.

Plugging in the definition of $V$ from Lemma 2.3, this readily yields the following ELBO of our generative model in (1). The variational gap can be found in Proposition 2.6 and Remark A.6.

**Corollary 2.5** (Evidence lower bound). *For every $u \in \mathcal{U}$ it holds almost surely that*

$$\log p_{X_T}(Y_0^u) \geq \mathbb{E}\left[\log p_{X_0}(Y_T^u) - \mathcal{R}_\mu^u(Y^u)\big|Y_0^u\right], \tag{10}$$

*where equality is obtained for $u^* \coloneqq \sigma^\top \nabla \log \breve{p}_X$.*

Comparing (8) and (10), we see that the ELBO equals the negative control costs. With the initial condition $Y_0^u \sim \mathcal{D}$, it represents a lower bound on the negative log-likelihood of our generative model. In practice, one can now parametrize $u$ with, for instance, a neural network and rely on gradient-based optimization to maximize the ELBO using samples from $\mathcal{D}$. The optimality condition in Corollary 2.5 guarantees that $\breve{p}_X = p_{Y^{u^*}}$ almost everywhere if we ensure that $X_0 \sim Y_T^{u^*}$, see Section A.3. In particular, this implies that $X_T \sim \mathcal{D}$, i.e., our generative model solves Problem 2.1. However, we still face the problem of sampling $X_0 \sim Y_T^{u^*}$ since the distribution of $Y_T^{u^*}$ depends[5] on the initial distribution $\mathcal{D}$. In Sections 2.4 and 3 we will demonstrate ways to circumvent this problem.

## 2.3 Path space perspective: KL divergence in continuous time

In this section, we show that the variational gap corresponding to Corollary 2.5 can be interpreted as a KL divergence between measures on the space of continuous trajectories, also known as *path space* (Üstünel & Zakai, 2013). Later, we can use this result to develop our sampling method. To this end, let us define the path space measure $\mathbb{P}_{Y^u}$ as the distribution of the trajectories associated with the controlled process $Y^u$ as defined in (7), see also Section A.1. Consequently, we denote by $\mathbb{P}_{Y^0}$ the path space measure associated with the uncontrolled process $Y^0$ with the choice $u \equiv 0$.

We can now state a path space measure perspective on Problem 2.1 by identifying a formula for the target measure $\mathbb{P}_{Y^{u^*}}$, which corresponds to the process in (7) with the optimal control $u = u^* = \sigma^\top \nabla \log \breve{p}_X$. We further show that, with the correct initial condition $Y_0^u \sim X_T$, this target measure corresponds to the measure of the time-reversed process, i.e., $\mathbb{P}_{Y^{u^*}} = \mathbb{P}_{\breve{X}}$, see Remark 2.2 and Section A.3. For the proof and further details, see Proposition A.9 in the appendix.

**Proposition 2.6** (Optimal path space measure). *The optimal path space measure can be defined via the work functional $\mathcal{W}: C([0,T], \mathbb{R}^d) \to \mathbb{R}$ and the Radon-Nikodym derivative*

$$\frac{d\mathbb{P}_{Y^{u^*}}}{d\mathbb{P}_{Y^0}} = \exp\left(-\mathcal{W}\right) \quad with \quad \mathcal{W}(Y^0) \coloneqq \mathcal{R}_\mu^0(Y^0) - \log \frac{p_{X_0}\left(Y_T^0\right)}{p_{X_T}\left(Y_0^0\right)}, \tag{11}$$

*where $\mathcal{R}_\mu^0(Y^0)$ is as in (9) with $u = 0$. Moreover, for any $u \in \mathcal{U}$, the expected variational gap*

$$G(u) \coloneqq \mathbb{E}\left[\log p_{X_T}(Y_0^u)\right] - \mathbb{E}\left[\log p_{X_0}(Y_T^u) - \mathcal{R}_\mu^u(Y^u)\right] \tag{12}$$

*of the ELBO in Corollary 2.5 satisfies that*

$$G(u) = D_{\mathrm{KL}}(\mathbb{P}_{Y^u}|\mathbb{P}_{Y^{u^*}}) = D_{\mathrm{KL}}(\mathbb{P}_{Y^u}|\mathbb{P}_{\breve{X}}) - D_{\mathrm{KL}}(\mathbb{P}_{Y_0^u}|\mathbb{P}_{X_T}). \tag{13}$$

Note that the optimal change of measure (11) can be seen as a version of *Doob's h-transform*, see Dai Pra & Pavon (1990). Furthermore, it can be interpreted as Bayes' rule for conditional probabilities, with the target $\mathbb{P}_{Y^{u^*}}$ denoting the posterior, $\mathbb{P}_{Y^0}$ the prior measure, and $\exp\left(-\mathcal{W}\right)$ being the likelihood. Formula (13) emphasizes again that we can solve Problem 2.1 by approximating the optimal control $u^*$ and having matching initial conditions, see also Remark 2.2. The next section provides an objective for this goal that can be used in practice.

---

[5]In case one has access to samples from the data distribution $\mathcal{D}$, one could use these as initial data $Y_0^{u^*}$ in order to simulate $X_0 \sim Y_T^{u^*}$. In doing so, however, one cannot expect to recover the entire distribution $\mathcal{D}$, but only the empirical distribution of the samples.

## 2.4 Connection to denoising score matching objective

This section outlines that, under a reparametrization of the generative model in (1), the ELBO in Corollary 2.5 corresponds to the objective typically used for the training of continuous-time diffusion models. We note that the ELBO in Corollary 2.5 in fact equals the one derived in Huang et al. (2021, Theorem 3). Following the arguments therein and motivated by Remark 2.2, we can now use the reparametrization $\mu := \sigma u - f$ to arrive at an uncontrolled *inference SDE* and a controlled *generative SDE*

$$\mathrm{d}Y_s = f(Y_s, s)\,\mathrm{d}s + \sigma(s)\,\mathrm{d}B_s, \quad Y_0 \sim \mathcal{D}, \quad \text{and} \quad \mathrm{d}X_s^u = \left(\bar{\sigma}\bar{u} - \overleftarrow{f}\right)(X_s^u, s)\,\mathrm{d}s + \bar{\sigma}(s)\,\mathrm{d}B_s. \tag{14}$$

In practice, the coefficients $f$ and $\sigma$ are usually[6] constructed such that $Y$ is an Ornstein-Uhlenbeck (OU) process with $Y_T$ being approximately distributed according to a standard normal distribution, see also (139) in Section A.13. This is why the process $Y$ is said to *diffuse* the data. Setting $X_0^u \sim \mathcal{N}(0, \mathrm{I})$ thus satisfies that $p_{X_0^u} \approx p_{Y_T}$ and allows to easily sample $X_0^u$.

The corresponding ELBO in Corollary 2.5 now takes the form

$$\log p_{X_T^u}(Y_0) \geq \mathbb{E}\left[\log p_{X_0^u}(Y_T) - \mathcal{R}_{\sigma u - f}^u(Y)\big|Y_0\right], \tag{15}$$

where, in analogy to (13), the expected variational gap is given by the forward KL divergence $G(u) = D_{\mathrm{KL}}(\mathbb{P}_Y|\mathbb{P}_{\bar{X}^u}) - D_{\mathrm{KL}}(\mathbb{P}_{Y_0}|\mathbb{P}_{X_T^u})$, see Proposition A.10. We note that the process $Y$ in the ELBO does not depend on the control $u$ anymore. Under suitable assumptions, this allows us to rewrite the expected negative ELBO (up to a constant not depending on $u$) as a denoising score matching objective (Vincent, 2011), i.e.,

$$\mathcal{L}_{\mathrm{DSM}}(u) := \frac{T}{2}\mathbb{E}\left[\left\|u(Y_\tau, \tau) - \sigma^\top(\tau)\nabla \log p_{Y_\tau|Y_0}(Y_\tau|Y_0)\right\|^2\right], \tag{16}$$

where $\tau \sim \mathcal{U}([0, T])$ and $p_{Y_\tau|Y_0}$ denotes the conditional density of $Y_\tau$ given $Y_0$, see Section A.9. We emphasize that the conditional density can be explicitly computed for the OU process, see also (137) in Section A.13. Due to its simplicity, variants of the objective (16) are typically used in implementations. Note, however, that the setting in this section requires that one has access to samples of $\mathcal{D}$ in order to simulate the process $Y$. In the next section, we consider a different scenario, where instead we only have access to the (unnormalized) density of $\mathcal{D}$.

## 3 Sampling from unnormalized densities

In many practical settings, for instance, in Bayesian statistics or computational physics, the data distribution $\mathcal{D}$ admits the density $\rho/\mathcal{Z}$, where $\rho$ is known, but computing the normalizing constant $\mathcal{Z} := \int_{\mathbb{R}^d} \rho(x)\,\mathrm{d}x$ is intractable and samples from $\mathcal{D}$ are not easily available. In this section, we propose a novel method based on diffusion models, called *time-reversed diffusion sampler* (DIS), which allows to sample from $\mathcal{D}$, see Figure 3. To this end, we interchange the roles of $X$ and $Y^u$ in our derivation in Section 2, i.e., consider

$$\mathrm{d}Y_s = f(Y_s, s)\,\mathrm{d}s + \sigma(s)\,\mathrm{d}B_s, \quad Y_0 \sim \mathcal{D}, \quad \text{and} \quad \mathrm{d}X_s^u = \left(\bar{\sigma}\bar{u} - \overleftarrow{f}\right)(X_s^u, s)\,\mathrm{d}s + \bar{\sigma}(s)\,\mathrm{d}B_s, \tag{17}$$

where we also renamed $\mu$ to $f$ to stay consistent with (14). Analogously to Theorem 2.4, the following corollary specifies the control objective, see Corollary A.12 in the appendix for details and the proof.

**Corollary 3.1** (Reverse KL divergence). *Let $X^u$ and $Y$ be defined by* (17). *Then it holds*

$$D_{\mathrm{KL}}(\mathbb{P}_{X^u}|\mathbb{P}_{\bar{Y}}) = D_{\mathrm{KL}}(\mathbb{P}_{X^u}|\mathbb{P}_{X^{u*}}) + D_{\mathrm{KL}}(\mathbb{P}_{X_0^u}|\mathbb{P}_{Y_T}) \tag{18a}$$

$$= \mathbb{E}\left[\mathcal{R}_{\bar{f}}^{\bar{u}}(X^u) + \log \frac{p_{Y_T}(X_0^u)}{\rho(X_T^u)}\right] + \log \mathcal{Z} + D_{\mathrm{KL}}(\mathbb{P}_{X_0^u}|\mathbb{P}_{Y_T}) \tag{18b}$$

$$= \mathcal{L}_{\mathrm{DIS}}(u) + \log \mathcal{Z}, \tag{18c}$$

---

[6]For coefficients typically used in practice (leading, for instance, to continuous-time analogs of SMLD and DDPM) we refer to Song et al. (2021) and Section A.13.

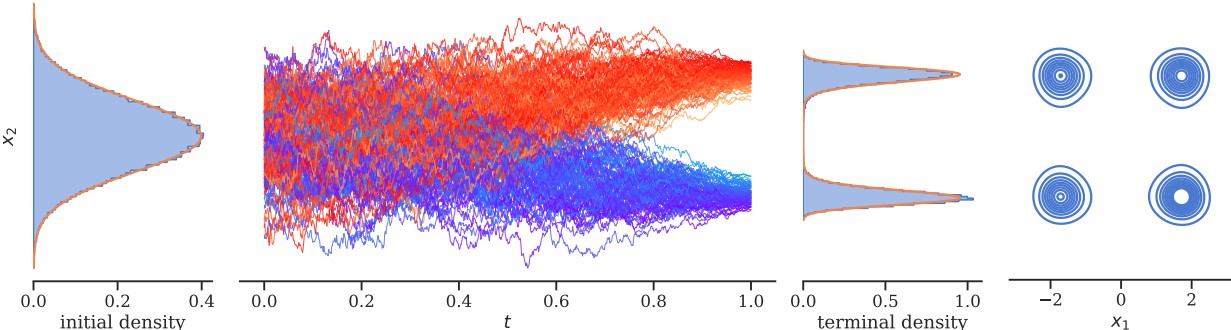

Figure 3: Illustration of our DIS algorithm for the double well example in Section 4 with $d = 20$, $w = 5$, $\delta = 3$. The process $X^u$ starts from a Gaussian (approximately distributed as $Y_T$) and the control $u$ is trained such that the distribution at terminal time $X_T^u$ approximates the target density $\rho/\mathcal{Z}$. The plot displays some trajectories as well as histograms at initial and terminal times. In the right panel, we show a KDE density estimation of a $2d$ marginal of the corresponding double well.

*where*

$$\mathcal{L}_{\mathrm{DIS}}(u) := \mathbb{E}\left[\mathcal{R}_{\bar{f}}^{\bar{u}}(X^u) + \log\frac{p_{X_0^u}(X_0^u)}{\rho(X_T^u)}\right]. \tag{19}$$

*In particular, this implies that*

$$D_{\mathrm{KL}}(\mathbb{P}_{X_0^u}|\mathbb{P}_{Y_T}) - \log\mathcal{Z} = \min_{u\in\mathcal{U}}\mathcal{L}_{\mathrm{DIS}}(u) \tag{20}$$

*and, assuming that $X_0^u \sim Y_T$, the minimizing control $u^* := \sigma^\top\nabla\log p_Y$ guarantees that $X_T^{u^*} \sim \mathcal{D}$.*

As in the previous chapter, $X_0^u \sim Y_T$ can be approximately achieved by choosing $X_0^u \sim \mathcal{N}(0, \mathrm{I})$ and $f$ and $\sigma$ such that $p_{Y_T} \approx \mathcal{N}(0, \mathrm{I})$, which incurs an irreducible prior loss given by $D_{\mathrm{KL}}(\mathbb{P}_{X_0^u}|\mathbb{P}_{Y_T})$. In practice, one can now minimize the control objective (19) using gradient-based optimization techniques. Alternatively, one can solve the corresponding HJB equation to obtain an approximation to $u^*$, see Section A.11. Note that, in contrast to (15) or (16), we do not need access to samples $Y_0 \sim \mathcal{D}$ as $Y$ does not need to be simulated for the objective in (19). However, since the controlled process $X^u$ now appears in the objective, we need to simulate the whole trajectory of $X^u$ and cannot resort to a Monte Carlo approximation, such as in the denoising score matching objective in (16). In Section A.14, we show that our path space perspective allows us to introduce divergences different from the reverse KL-divergence in (18a), which allows for off-policy training and can result in improved numerical performance.

### 3.1 Comparison to Schrödinger half-bridges

The idea to rely on controlled diffusions in order to sample from prescribed target densities is closely related to Schrödinger bridges (Dai Pra, 1991). This problem has been particularly well studied in the case where the initial density follows a Dirac distribution, i.e., the stochastic process starts at a pre-specified point—often referred to as Schrödinger half-bridge. Corresponding numerical algorithms based on deep learning, referred to as *Path Integral Sampler* (PIS) in Zhang & Chen (2022a), have been independently presented in Richter (2021); Zhang & Chen (2022a); Vargas et al. (2023b). Combining ideas from Dai Pra (1991) and Fleming & Soner (2006), it can be shown that a corresponding control problem can be formulated as

$$-\log\mathcal{Z} = \min_{u\in\mathcal{U}}\mathcal{L}_{\mathrm{PIS}}(u), \quad \text{with} \quad \mathcal{L}_{\mathrm{PIS}}(u) := \mathbb{E}\left[\mathcal{R}_0^{\bar{u}}(X^u) + \log\frac{p_{X_T^0}(X_T^u)}{\rho(X_T^u)}\right], \tag{21}$$

where the controlled diffusion $X^u$ is defined as in (17) with a fixed initial value $X_0^u = x_0 \in \mathbb{R}^d$. In the above, $p_{X_T^0}$ denotes the density of the uncontrolled process $X_T^0$ at time $T$. Equivalently, one can show that $\mathbb{P}_{X^{u^*}}$

satisfies the optimal change of measure, given by

$$\frac{d\mathbb{P}_{X^{u^*}}}{d\mathbb{P}_{X^0}}(X) = \frac{\rho}{\mathcal{Z} p_{X^0_T}}(X_T), \tag{22}$$

for all suitable stochastic processes $X$ on the path space. Using the Girsanov theorem (Theorem A.7), we see that

$$D_{\mathrm{KL}}(\mathbb{P}_{X^u}|\mathbb{P}_{X^{u^*}}) = \mathcal{L}_{\mathrm{PIS}}(u) + \log \mathcal{Z}, \tag{23}$$

see also Zhang & Chen (2022a); Nüsken & Richter (2021). Similar to Corollary 3.1, one can thus show that the optimally controlled process satisfies $X^{u^*}_T \sim \mathcal{D}$, see also Tzen & Raginsky (2019) and Pavon (2022). Comparing the two attempts, i.e., the objectives (19) and (21), we can identify multiple differences:

- Different initial distributions, running costs, and terminal costs are considered. In the Schrödinger half-bridge, the terminal costs consist of $\rho$ and $p_{X^0_T}$, whereas for the diffusion model they only consist of $\rho$.

- For the Schrödinger half-bridge, $f$ needs to be chosen such that $p_{X^0_T}$ is known analytically (up to a constant). For the time-reversed diffusion sampler, $f$ needs to be chosen such that $p_{Y_T} \approx p_{X^u_0}$ in order to have a small prior loss $D_{\mathrm{KL}}(\mathbb{P}_{X^u_0}|\mathbb{P}_{Y_T})$.

- In the Schrödinger half-bridge, $X^u_0$ starts from an arbitrary, but fixed, point, whereas for the diffusion-based generative modeling attempt $X^u_0$ must be (approximately) distributed as $Y_T$.

In Appendix A.10, we show that the optimal control $u^*$ of our objective in (19) can be numerically more stable than the one in (21). In the next section, we demonstrates that this can also lead to better sample quality and more accurate estimates of normalizing constants.

## 4 Numerical examples

The numerical experiments displayed in Figures 4 and 5 show that our *time-reversed diffusion sampler* (DIS) succeeds in sampling from high-dimensional multimodal distributions. We compare our method against the *Path Integral Sampler* (PIS) introduced in Zhang & Chen (2022a), which uses the objective from Section 3.1. As shown in Zhang & Chen (2022a), the latter method can already outperform various state-of-the-art sampling methods. This includes gradient-guided MCMC methods without the annealing trick, such as Hamiltonian Monte Carlo (HMC) (MacKay, 2003) and No-U-Turn Sampler (NUTS) (Hoffman & Gelman, 2014), SMC with annealing trick such as Annealed Flow Transport Monte Carlo (AFT) (Arbel et al., 2021), and variational normalizing flows (VINF) (Rezende & Mohamed, 2015).

Let us summarize our setting in the following and note that further details, as well as the exact hyperparameters, can be found in Section A.13 and Table 2. Our PyTorch implementation[7] is based on Zhang & Chen (2022a) with the following main differences: We train with larger batch sizes and more gradient steps to guarantee better convergence. We further use a clipping schedule for the neural network outputs, and an exponential moving average of the parameters for evaluation (as is often done for diffusion models (Nichol & Dhariwal, 2021)). Moreover, we train with a varying number of step sizes for simulating the SDEs using the Euler-Maruyama (EM) scheme in order to amortize the cost of training separate models for different evaluation settings. To have a fair comparison, we trained PIS using the same setup. We also present a comparison to the original code in Figure 11, showing that our setup leads to increased performance and stability. The optimization routine is summarized in Algorithm 1.

Similar to PIS, we also use the score of the density $\nabla \log \rho$ (typically given in closed-form or evaluated via automatic differentiation) for our parametrization of the control $u_\theta$, given by

$$u_\theta(x, t) := \Phi^{(1)}_\theta(x, t) + \Phi^{(2)}_\theta(t)\sigma(t)s(x, t). \tag{24}$$

Here, $s$ is a linear interpolation between the initial and terminal scores $\nabla \log \rho$ and $\nabla \log p_{X^u_0}$, thus approximating the optimal score $\nabla \log p_Y$ at the boundary values, see Corollary 3.1. This parametrization yielded

---

[7]The associated repository can be found at `https://github.com/juliusberner/sde_sampler`.

---

**Algorithm 1** Time-reversed diffusion sampler (DIS): Training the control in Section 3 via deep learning.

---

**input** neural network $u_\theta$ with initial parameters $\theta^{(0)}$, optimizer method step for updating the parameters, number of steps $K$, batch size $m$
**output** parameters $(\theta^{(k)})_{k=1}^K$
    **for** $k \leftarrow 0, \ldots, K-1$ **do**
        $(x^{(i)})_{i=1}^m \leftarrow$ sample from $\mathcal{N}(0, \mathrm{I})^{\otimes m}$
        $\widehat{\mathcal{L}}_{\mathrm{DIS}}(u_{\theta^{(k)}}) \leftarrow$ estimate the cost in (19) using the EM scheme with $\widehat{X}_0^\theta = x^{(i)}$, $i = 1, \ldots, m$
        $\theta^{(k+1)} \leftarrow \mathrm{step}\left(\theta^{(k)}, \nabla\widehat{\mathcal{L}}_{\mathrm{DIS}}(u_{\theta^{(k)}})\right)$
    **end for**

---

the best results in practice, see also the comparison to other parametrizations in Section A.14. For the drift and diffusion coefficients, we can make use of successful noise schedules for diffusion models and choose the *variance-preserving* SDE from Song et al. (2021).

We evaluate DIS and PIS on a number of examples that we outline in the sequel. Specifically, we compare the approximation of the log-normalizing constant $\log \mathcal{Z}$ using (approximately) unbiased estimators given by importance sampling in path space, see Section A.12 for DIS and Zhang & Chen (2022a) for PIS. To evaluate the sample quality, we compare Monte Carlo approximations of expectations as well as estimates of standard deviations. We further refer to Figure 7 in the appendix for preliminary experiments using *Physics informed neural networks* (PINNs) for solving the corresponding HJB equation, as well as to Figure 6, where we employ a divergence different from the KL divergence, which is possible due to our path space perspective.

### 4.1 Examples

Let us present the numerical examples on which we evaluate our method.

**Gaussian mixture model (GMM):** We consider

$$\rho(x) = \sum_{m=1}^M \alpha_m \mathcal{N}(x; \widetilde{\mu}_m, \Sigma_m), \tag{25}$$

with $\sum_{m=1}^M \alpha_m = 1$. Specifically, we choose $m = 9$, $\Sigma_m = 0.3\,\mathrm{I}$, and $(\widetilde{\mu}_m)_{m=1}^9 = \{-5, 0, 5\} \times \{-5, 0, 5\} \subset \mathbb{R}^2$. The density and the optimal drift are depicted in Figure 2.

**Funnel:** The 10-dimensional *Funnel distribution* (Neal, 2003) is a challenging example often used to test MCMC methods. It is given by

$$\rho(x) = \mathcal{N}(x_1; 0, \nu^2) \prod_{i=2}^d \mathcal{N}(x_i; 0, e^{x_1}) \tag{26}$$

for $x = (x_i)_{i=1}^{10} \in \mathbb{R}^{10}$ with[8] $\nu = 3$.

**Double well (DW):** A typical problem in molecular dynamics considers sampling from the stationary distribution of a Langevin dynamics, where the drift of the SDE is given by the negative gradient of a potential $\Psi$, namely

$$\mathrm{d}X_s = -\nabla\Psi(X_s)\,\mathrm{d}s + \sigma(s)\,\mathrm{d}B_s, \tag{27}$$

see, e.g., (Leimkuhler & Matthews, 2015). Given certain assumptions on the potential, the stationary density of the process can be shown to be $p_{X_\infty} = e^{-\Psi}/\mathcal{Z}$. Potentials often contain multiple *energy barriers*, and resulting local minima correspond to meaningful configurations of a molecule. However, the convergence speed of $p_{X_t}$ can be very slow – in particular for large energy barriers. Our time-reversed diffusion sampler,

---

[8]Due to a typo in the code, the results presented by Zhang & Chen (2022a) seem to consider $\nu = 3$ for the baselines, but the favorable choice of $\nu = 1$ for evaluating their method.

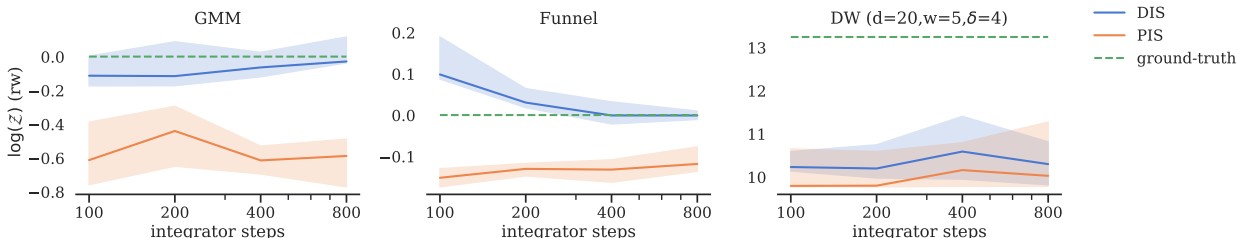

Figure 4: We compare our DIS method against PIS on the ability to compute the log-normalizing constant $\log \mathcal{Z}$ (median and interquartile range over 10 training seeds) when using $N \in \{100, 200, 400, 800\}$ steps of the Euler-Maruyama scheme. Our method outperforms PIS clearly for the GMM and Funnel examples and offers a slight improvement for the DW example. Each model has been trained with each $N$ for $1/4$ of the total gradient steps (starting with 100 and ending with 800 steps). See also Figure 10 for a comparison of models trained on a single step size.

on the other hand, can (at least in principle) sample from densities already at a fixed time $T$. In our example, we shall consider a $d$-dimensional *double well* potential, corresponding to an (unnormalized) density $\rho = e^{-\Psi}$ given by

$$\rho(x) = \exp\left(-\sum_{i=1}^{w}(x_i^2 - \delta)^2 - \frac{1}{2}\sum_{i=w+1}^{d} x_i^2\right) \tag{28}$$

with $w \in \mathbb{N}$ combined double wells and a separation parameter $\delta \in (0, \infty)$, see also Wu et al. (2020). Note that, due to the double well structure of the potential, the density contains $2^w$ modes. For these multimodal examples, we can compute a reference solution of the log-normalizing constant $\log \mathcal{Z}$ and other statistics of interest by numerical integration since $\rho$ factorizes in the dimensions.

### 4.2 Results

Our experiments show that DIS can offer improvements over PIS for all the tasks we considered, i.e., estimation of normalizing constants, expectations, and standard deviations. Figures 4 and 5 display direct comparisons for the examples in Section 4.1, noting that we use the same training techniques but different objectives for the respective methods. Moreover, we recall that our setup of the PIS method already outperforms the default implementation by Zhang & Chen (2022a), see Figure 11 in the appendix. We also emphasize that the reference process of PIS satisfies that $X_T^0 \sim \mathcal{N}(0, \mathrm{I})$, which is already the correct distribution for $d - w$ dimensions of the double well example. Nevertheless, our proposed DIS method converges to a better approximation and provides strong results even for multimodal, high-dimensional distributions, such as $d = 50$ with 32 well-separated modes, see Figure 5. Finally, we provide further results for two promising future directions: We show that replacing the KL divergence in Corollary 3.1 with the *log-variance divergence* (Nüsken & Richter, 2021; Richter et al., 2020) can further improve performance, see Figure 6 and Section A.14 for details. In Figure 7, we show that solving the HJB equation in Lemma 2.3 using *physics-informed neural networks* (PINNs) can provide competitive results, see Section A.11 for further explanations. We present more numerical results and comparisons in Sections A.13 and A.14.

### 4.3 Limitations

In this section we shall discuss limitations of the optimal control perspective and our resulting sampling algorithm. First, note that the control perspective so far only yields new algorithms if the target density is known (up to the normalization constant). In particular, we note that the direct minimization of divergences, such as the KL or log-variance divergence, needs access to the (unnormalized) target density. This is typically not the case for problems in generative modeling where one has only access to data samples and where one resorts to optimizing the ELBO, as outlined in Section 2.4. However, we emphasize that classical sampling

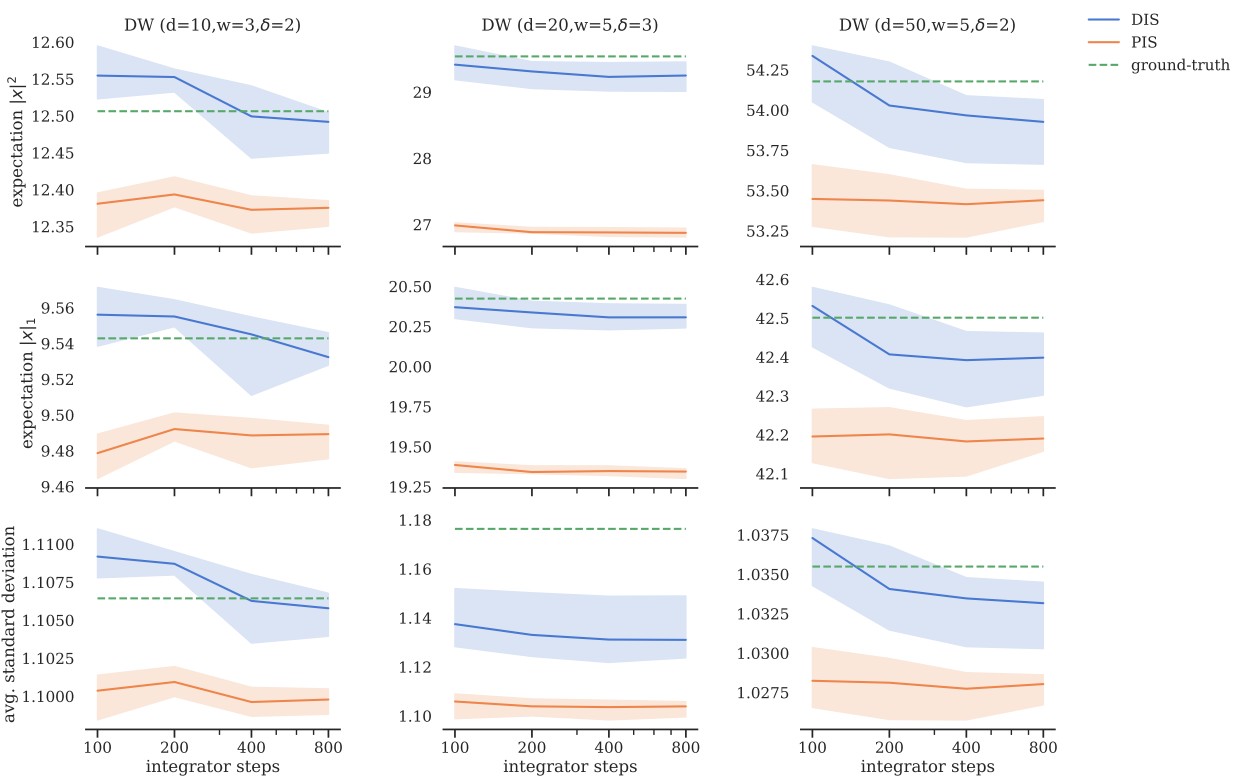

Figure 5: We compare our DIS method against PIS on the ability to estimate the expectations $\mathbb{E}\left[\|Y_0\|^2\right]$ and $\mathbb{E}\left[\|Y_0\|_1\right]$ and the average standard deviation $\frac{1}{d}\sum_{i=1}^{d}\sqrt{\mathbb{V}\left[(Y_0)_i\right]}$, see (148). Each model has been trained with each $N \in \{100, 200, 400, 800\}$ for $1/4$ of the total gradient steps (starting with 100 and ending with 800 steps). We compare the methods using $N$ steps of the Euler-Maruyama scheme when computing our estimates (median and interquartile range over 10 training seeds). Our method outperforms PIS in all considered settings in terms of accuracy.

problems (without any samples from the data distribution) can arguably be more challenging, e.g., due to exploration-exploitation trade-offs and potential mode collapse.

Compared to classical sampling methods such as, e.g., MCMC and SMC methods, the sampling time of our algorithm DIS is typically much faster. However, this comes at the cost of training a model first, which might only amortize when a larger number of samples is needed. While a well-trained model enjoys good sampling guarantees (cf. De Bortoli (2022); Chen et al. (2022); Lee et al. (2023)), approximating the optimal control (along the controlled dynamics) can be challenging and sensitive to the chosen hyperparameters. We have observed that clever choices of alternative divergences can improve convergence and counteract mode collapse, however, it is generally difficult to provide convergence guarantees.

## 5    Conclusion and outlook

We propose a connection of diffusion models to the fields of optimal control and reinforcement learning, which provides valuable new insights and allows to transfer established tools from one field to the respective other. As first steps, we have shown how to readily derive the ELBO for continuous-time diffusion models solely from control arguments, we provided an interpretation of diffusion models via measures on path space (eventually leading to improved losses), and we extended the diffusion modeling framework to sampling from unnormalized densities. We could demonstrate that our framework offers significant numerical advantages over existing diffusion-based sampling approaches across a series of challenging, high-dimensional problems.

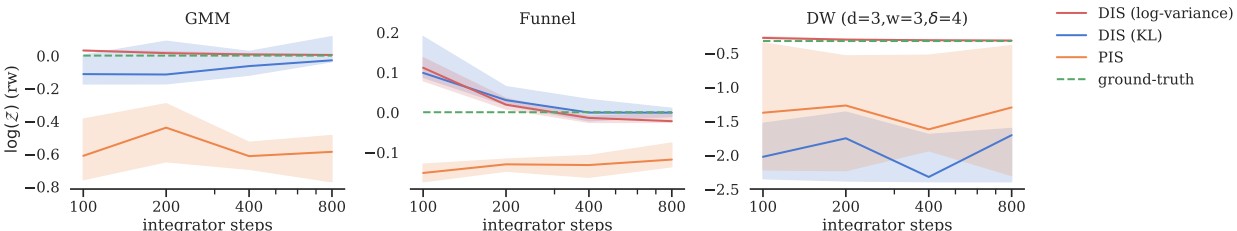

Figure 6: We use the settings in Figure 4 and only change the reverse KL divergence in Corollary 3.1 to the log-variance divergence (Nüsken & Richter, 2021). Training with the latter objective can significantly improve the estimate of the log-normalizing constant $\log \mathcal{Z}$ for the GMM, the Funnel, and a DW example.

With the new connections at hand, we anticipate further fruitful theoretical as well as practical insights. For instance, the PDE perspective offers to rely on PDE techniques and solvers that have been developed in the past, see Section A.11 for details. For high-dimensional settings, tensor-based methods (Richter et al., 2023), as well as SDE-based solvers (Nüsken & Richter, 2021), seem to be well suited for numerical approximation. Alternatively, one can readily apply numerical methods from control theory to approximate the score function, such as, for instance, methods based on tensor trains (Oster et al., 2022) or policy iteration, which has recently been combined with deep learning (Zhou et al., 2021).

Note that one usually controls linear, Ornstein-Uhlenbeck type processes. This motivates to approximate the target density by a Gaussian density (which corresponds to quadratic costs in the control problem). Then, one can pretrain the score model by solving a linear-quadratic-control problem, for which very efficient numerical methods have been developed extensively. On the other hand, we may add additional running costs to the control objective in order to promote or prevent certain regions in the domain, which can incorporate domain knowledge and improve sampling for respective applications. Interesting future perspectives also include the usage of other divergences on path space (motivated by our promising results with the log-variance divergence), as well as the extension to Schrödinger bridges, for instance based on the work by Chen et al. (2021). These methods are particularly well-suited for the task of sampling from unnormalized densities, where we cannot leverage the efficient denoising score matching objective.

## Acknowledgments

We would like to thank Nikolas Nüsken and Ricky T. Q. Chen for many useful discussions. The research of Lorenz Richter has been partially funded by Deutsche Forschungsgemeinschaft (DFG) through the grant CRC 1114 "Scaling Cascades in Complex Systems" (project A05, project number 235221301). Julius Berner is grateful to G-Research for the travel grant.

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

# A   Appendix

## Contents

## A.1   Setting

Let $d, k \in \mathbb{N}$ and $T \in (0, \infty)$. For a random variable $X$, which is absolutely continuous w.r.t. to the Lebesgue measure, we write $p_X$ for its density. We denote by $B$ a standard $d$-dimensional Brownian motion. We say that a continuous $\mathbb{R}^d$-valued stochastic process $Y = (Y_t)_{t \in [0,T]}$ has density $p_Y \colon \mathbb{R}^d \times [0, T] \to [0, \infty)$ if for all $t \in [0, T]$ the random variable $Y_t$ has density $p_Y(\cdot, t)$ w.r.t. to the $d$-dimensional Lebesgue measure, i.e., for all $t \in [0, T]$ and all measurable $A \subset \mathbb{R}^d$ it holds that

$$\mathbb{P}\left[Y_t \in A\right] = \int_A p_Y(x, t)\, \mathrm{d}x = \int_A p_{Y_t}(x)\, \mathrm{d}x. \tag{29}$$

We denote by $\mathbb{P}_Y$ the law of $Y$ on the space of continuous functions $C([0, T], \mathbb{R}^d)$ equipped with the Borel measure. We assume that the coefficient functions and initial conditions of all appearing SDEs are sufficiently regular such that the SDEs admit unique strong solutions and Novikov's condition is satisfied, see, e.g., Øksendal & Øksendal (2003, Section 8.6). Furthermore, we assume that the SDE solutions have densities that are smooth and strictly positive for $t \in (0, T)$ and that can be written as unique solutions to corresponding Fokker-Planck equations, see, for instance, Arnold (1974, Section 2.6) and Baldi (2017, Section 10.5) for the details. For a function $f \colon \mathbb{R}^d \times [0, T] \to \mathbb{R}^k$, we write $\overleftarrow{f}$ for the time-reversed function given by

$$\overleftarrow{f}(x, t) = f(x, T - t), \quad (x, t) \in \mathbb{R}^d \times [0, T]. \tag{30}$$

For a scalar-valued function $g \colon \mathbb{R}^d \times [0, T] \to \mathbb{R}$, we denote by $\nabla g$, $\nabla^2 g$, and $\mathrm{div}(g)$ its gradient, Hessian matrix, and divergence w.r.t. to the spatial variable $x$. For matrix-valued functions $A \colon \mathbb{R}^d \to \mathbb{R}^{d \times d}$, we denote

by

$$\operatorname{Tr}(A) := \sum_{i=1}^{d} A_{ii} \tag{31}$$

the trace of its output. Finally, we define the divergence of matrix-valued functions row-wise, see Section A.2.

## A.2  Identities for divergences

Let $A \colon \mathbb{R}^d \to \mathbb{R}^{d \times d}$, $v \colon \mathbb{R}^d \to \mathbb{R}^d$, and $g \colon \mathbb{R}^d \to \mathbb{R}$. We define the divergence of $A$ row-wise, i.e.,

$$\operatorname{div}(A) := (\operatorname{div}(A_{i\cdot}))_{i=1}^{d} = \sum_{j=1}^{d} \partial_{x_j} A_{\cdot j}, \tag{32}$$

where $A_{i\cdot}$ and $A_{\cdot j}$ denote the $i$-th row and $j$-th column, respectively. Then the following identities hold true:

1. $\operatorname{div}(\operatorname{div}(A)) = \sum_{i,j=1}^{d} \partial_{x_i} \partial_{x_j} A_{ij}$

2. $\operatorname{div}(vg) = \operatorname{div}(v)g + v \cdot \nabla g$

3. $\operatorname{div}(Ag) = \operatorname{div}(A)g + A \nabla g$

4. $\operatorname{div}(Av) = \operatorname{div}(A^\top) \cdot v + \operatorname{Tr}(A \nabla v)$.

## A.3  Reverse-time SDEs

The next theorem shows that the marginals of a time-reversed Itô process can be represented as marginals of another Itô process, see Huang et al. (2021); Song et al. (2021); Nelson (1967); Anderson (1982); Haussmann & Pardoux (1986); Föllmer (1988). We present a formulation from Huang et al. (2021, Appendix G) which derives a whole family of processes (parametrized by a function $\lambda$). The relations stated in equation (2) follow from the choice $\lambda = 0$ and the fact that $\operatorname{div}(D) = 0$ if $\sigma$ does not depend on the spatial variable $x$.

**Theorem A.1** (Reverse-time SDE). *Let $f \in \mathbb{R}^d \times [0,T] \to \mathbb{R}^d$ and $\sigma \colon \mathbb{R}^d \times [0,T] \to \mathbb{R}^{d \times d}$, let $Y = (Y_s)_{s \in [0,T]}$ be the solution to the SDE*

$$\mathrm{d}Y_s = f(Y_s, s)\,\mathrm{d}s + \sigma(Y_s, s)\,\mathrm{d}B_s, \tag{33}$$

*and assume that $Y$ has density $p_Y$, which satisfies the Fokker-Planck equation given by*

$$\partial_t p_Y = \operatorname{div}\left(\operatorname{div}\left(D p_Y\right) - f p_Y\right), \tag{34}$$

*where $D := \frac{1}{2}\sigma\sigma^\top$. For every $\lambda \in C^2([0,T],[0,1])$ the solution $\overset{\leftarrow}{Y} = (\overset{\leftarrow}{Y}_s)_{s \in [0,T]}$ to the reverse-time SDE*

$$\mathrm{d}\overset{\leftarrow}{Y}_s = \overset{\leftarrow}{\mu}^{(\lambda)}(\overset{\leftarrow}{Y}_s, s)\,\mathrm{d}s + \overset{\leftarrow}{\sigma}^{(\lambda)}(\overset{\leftarrow}{Y}_s, s)\,\mathrm{d}B_s, \quad \overset{\leftarrow}{Y}_0 \sim Y_T, \tag{35}$$

*with*

$$\mu^{(\lambda)} := (2 - \lambda)\operatorname{div}\left(D\right) + (2 - \lambda)D \nabla \log p_Y - f, \tag{36}$$

*and*

$$\sigma^{(\lambda)} := \sqrt{1 - \lambda}\,\sigma \tag{37}$$

*has density $p_{\overset{\leftarrow}{Y}}$ given by*

$$p_{\overset{\leftarrow}{Y}}(\cdot, t) = \overset{\leftarrow}{p}_Y(\cdot, t) \tag{38}$$

*almost everywhere for every $t \in [0,T]$. In other words, for every $t \in [0,T]$ it holds that $Y_{T-t} \sim \overset{\leftarrow}{Y}_t$.*

*Proof.* Using the Fokker-Planck equation in (34), we observe that

$$\partial_t \overset{\leftarrow}{p}_Y = \operatorname{div}\left(-\operatorname{div}\left(\overset{\leftarrow}{D}\overset{\leftarrow}{p}_Y\right) + \overset{\leftarrow}{f}\overset{\leftarrow}{p}_Y\right). \tag{39}$$

The negative divergence, originating from the chain rule, prohibits us from directly viewing the above equation as a Fokker-Planck equation. We can, however, use the identities in Section A.2, to show that

$$\operatorname{div}\left(\breve{D}\breve{p}_Y\right) = \operatorname{div}\left(\breve{D}\right)\breve{p}_Y + \breve{D}\nabla\breve{p}_Y = \left(\operatorname{div}\left(\breve{D}\right) + \breve{D}\nabla\log\breve{p}_Y\right)\breve{p}_Y. \tag{40}$$

This implies that we can rewrite (39) as

$$\partial_t\breve{p}_Y = \operatorname{div}\left((1-\lambda)\operatorname{div}\left(\breve{D}\breve{p}_Y\right) - (2-\lambda)\operatorname{div}\left(\breve{D}\breve{p}_Y\right) + \breve{f}\breve{p}_Y\right) \tag{41a}$$

$$= \operatorname{div}\left(\operatorname{div}\left(\breve{D}^{(\lambda)}\breve{p}_Y\right) - \breve{\mu}^{(\lambda)}\breve{p}_Y\right), \tag{41b}$$

where $D^{(\lambda)} := \frac{1}{2}\sigma^{(\lambda)}(\sigma^{(\lambda)})^\top$. As the PDE in (41b) defines a valid Fokker-Planck equation associated to the reverse-time SDE given by (35), this proves the claim. $\qquad\square$

### A.4 Further details on the HJB equation

In order to solve Problem 2.1, one might be tempted to rely on classical methods to approximate the solution of the HJB equation from Lemma 2.3 directly. However, in the setting of Remark 2.2, one should note that the optimal drift,

$$\mu = \sigma\sigma^\top\nabla\log p_Y - f = -\sigma\sigma^\top\nabla V - f, \tag{42}$$

contains the solution $V$ itself. Plugging it into the HJB equation in Lemma 2.3, we get the equation

$$\partial_t V = \operatorname{Tr}\left(D\nabla^2 V\right) - f\cdot\nabla V + \operatorname{div}(f) - \frac{1}{2}\|\sigma^\top\nabla V\|^2, \quad V(\cdot,T) = -\log p_{X_0}. \tag{43}$$

Likewise, when applying the Hopf–Cole transformation from Section A.5 to $p_Y$ directly, i.e., considering $V := -\log p_Y$, where $Y$ is the solution to SDE (3), we get the same PDE. We note that the signs in (43) do not match with typical HJB equations from control theory. In order to obtain an HJB equation, we can consider the time-reversed function $\breve{V}$, which satisfies

$$\partial_t\breve{V} = -\operatorname{Tr}\left(\breve{D}\nabla^2\breve{V}\right) + \breve{f}\cdot\nabla\breve{V} - \operatorname{div}(\breve{f}) + \frac{1}{2}\|\breve{\sigma}^\top\nabla\breve{V}\|^2, \quad \breve{V}(\cdot,T) = -\log p_{X_T}. \tag{44}$$

Unfortunately, the terminal conditions in both (43) and (44) are typically not available in the context of generative modeling. Specifically, for the optimal drift in (42), they correspond to the intractable marginal densities of the inference process $Y$, since it holds that $p_X = \breve{p}_Y$, see Remark 2.2. However, the situation is different in the case of sampling from (unnormalized) densities, see Section 3.

### A.5 Hopf–Cole transformation

Recall that the Kolmogorov backward equation follows from the Fokker-Planck equation by using the divergence identities from Section A.2, i.e.

$$\partial_t\breve{p}_X = \operatorname{div}\left(-\operatorname{div}\left(D\breve{p}_X\right) + \mu\breve{p}_X\right) = \operatorname{div}\left(-D\nabla\breve{p}_X\right) + \mu\cdot\nabla\breve{p}_X + \operatorname{div}(\mu)\breve{p}_X \tag{45a}$$

$$= -\operatorname{Tr}\left(D\nabla^2\breve{p}_X\right) + \mu\cdot\nabla\breve{p}_X + \operatorname{div}(\mu)\breve{p}_X. \tag{45b}$$

The following lemma details the relation of the HJB equation in Lemma 2.3 and the linear Kolmogorov backward equation[9] in (5). A proof of the Hopf-Cole transformation can, e.g., be found in Evans (2010, Section 4.4.1) and Richter & Berner (2022, Appendix G). Lemma 2.3 follows with the choices $b := -\mu$ and $h := \operatorname{div}(\mu)$.

**Lemma A.2** (Hopf–Cole transformation). *Let $h\colon\mathbb{R}^d\times[0,T]\to\mathbb{R}$ and let $p\in C^{2,1}(\mathbb{R}^d\times[0,T],\mathbb{R})$ solve the linear PDE*

$$\partial_t p = -\frac{1}{2}\operatorname{Tr}(\sigma\sigma^\top\nabla^2 p) - b\cdot\nabla p + hp. \tag{46}$$

*Then $V := -\log p$ satisfies the HJB equation*

$$\partial_t V = -\frac{1}{2}\operatorname{Tr}(\sigma\sigma^\top\nabla^2 V) - b\cdot\nabla V - h + \frac{1}{2}\|\sigma^\top\nabla V\|^2. \tag{47}$$

---

[9]For $\operatorname{div}(\mu) = 0$ this can be viewed as the adjoint of the Fokker-Planck equation.

For further applications of the Hopf–Cole transformation we refer, for instance, to Fleming & Soner (2006); Hartmann et al. (2017); Léger & Li (2021).

### A.6    Brief introduction to stochastic optimal control

In this section, we shall provide a brief introduction to stochastic optimal control. For details and further reading, we refer the interested reader to the monographs by Fleming & Rishel (2012); Fleming & Soner (2006); Pham (2009); Van Handel (2007). Loosely speaking, stochastic control theory deals with identifying optimal strategies in noisy environments, in our case, continuous-time stochastic processes defined by the SDE

$$\mathrm{d}X_s^U = \widetilde{\mu}(X_s^U, s, U_s)\,\mathrm{d}s + \widetilde{\sigma}(X_s^U, s, U_s)\,\mathrm{d}B_s, \tag{48}$$

where $\widetilde{\mu}$ and $\widetilde{\sigma}$ are suitable functions, $B$ is a $d$-dimensional Brownian motion, and $U$ is a progressively measurable, $\mathbb{R}^d$-valued random control process. For ease of presentation, we focus on the frequent case where $U$ is a *Markov control*, which means that there exists a deterministic function $u \in \mathcal{U} \subset C(\mathbb{R}^d \times [0,T], \mathbb{R}^d)$, such that $U_s = u(X_s^U, s)$. In other words, the randomness of the process $U$ is only coming from the stochastic process $X^U$. The function class $\mathcal{U}$ then defines the set of *admissible controls*. Very often, one considers the special cases

$$\widetilde{\mu}(x, s, u) \coloneqq \mu(x, s) + \sigma(s)u(x, s) \qquad \text{and} \qquad \widetilde{\sigma}(x, s, u) \coloneqq \sigma(s), \tag{49}$$

where $\mu$ and $\sigma$ might correspond to the choices taken in Section 2.4, and one might think of $u$ as a steering force as to reach a certain target.

The goal is now to minimize specified control costs with respect to the control $u$. To this end, we can define the cost functional

$$J(u; x_{\mathrm{init}}, 0) = \mathbb{E}\left[\int_0^T \widetilde{h}(X_s^U, s, u(X_s^U, s))\,\mathrm{d}s + g(X_T^U)\,\bigg|\,X_0^U = x_{\mathrm{init}}\right], \tag{50}$$

where $\widetilde{h} : \mathbb{R}^d \times [0,T] \times \mathbb{R}^d \to \mathbb{R}$ specifies *running costs* and $g : \mathbb{R}^d \to \mathbb{R}$ represents *terminal costs*. Furthermore we can define the *cost-to-go* as

$$J(u; x, t) = \mathbb{E}\left[\int_t^T \widetilde{h}(X_s^U, s, u(X_s^U, s))\,\mathrm{d}s + g(X_T^U)\,\bigg|\,X_t^U = x\right], \tag{51}$$

now depending on respective initial values $(x, t) \in \mathbb{R}^d \times [0,T]$. The objective in optimal control is now to minimize this quantity over all admissible controls $u \in \mathcal{U}$ and we, therefore, introduce the so-called *value function*

$$V(x, t) = \inf_{u \in \mathcal{U}} J(u; x, t) \tag{52}$$

as the optimal costs conditioned on being in position $x$ at time $t$.

Motivated by the *dynamic programming principle* (Bellman, 1957), one can then derive the main result from control theory, namely that the function $V$ defined in (52) fulfills a nonlinear PDE, which can thus be interpreted as the determining equation for optimality[10].

**Theorem A.3** (Verification theorem for general HJB equation)**.** *Let $V \in C^{2,1}(\mathbb{R}^d \times [0,T], \mathbb{R})$ fulfill the PDE*

$$\partial_t V = -\inf_{\alpha \in \mathbb{R}^d}\left\{\widetilde{h}(\cdot, \cdot, \alpha) + \widetilde{\mu}(\cdot, \cdot, \alpha) \cdot \nabla V + \frac{1}{2}\operatorname{Tr}\left((\widetilde{\sigma}\widetilde{\sigma}^\top)(\cdot, \cdot, \alpha)\nabla^2 V\right)\right\}, \quad V(\cdot, T) = g, \tag{53}$$

*such that*

$$\sup_{(x,t)\in\mathbb{R}^d\times[0,T]} \frac{\|V(x,t)\|}{1 + \|x\|^2} < \infty \tag{54}$$

---

[10]In practice, solutions to optimal control problems may not posses enough regularity in order to formally fulfill the HJB equation, such that a complete theory of optimal control needs to introduce an appropriate concept of weak solutions, leading to so-called viscosity solutions that have been studied, for instance, in Fleming & Soner (2006); Lions (1983).

*and assume that there exists a measurable function $\mathcal{U} \ni u^* : \mathbb{R}^d \times [0, T] \to \mathbb{R}^d$ that attains the above infimum for all $(x, t) \in \mathbb{R}^d \times [0, T]$. Further, let the correspondingly controlled SDE in (48) with $U_s^* := u^*(X_s^{U^*}, s)$ have a strong solution $X^{U^*}$. Then $V$ coincides with the value function as defined in (52) and $u^*$ is an optimal Markovian control.*

Let us appreciate the fact that the infimum in the HJB equation in (53) is merely over the set $\mathbb{R}^d$ and not over the function space $\mathcal{U}$ as in (52), so the minimization reduces to a pointwise operation. A proof of Theorem A.3 can, for instance, be found in Pham (2009, Theorem 3.5.2).

In many applications, in addition to the choices (49), one considers the special form of running costs

$$\widetilde{h}(x, s, u(x, s)) := h(x, s) + \frac{1}{2}\|u(x, s)\|^2, \tag{55}$$

where $h : \mathbb{R}^d \times [0, T] \to \mathbb{R}^d$. In this setting, the minimization appearing in the general HJB equation (53) can be solved explicitly, therefore leading to a closed-form PDE, as made precise with the following Corollary.

**Corollary A.4** (HJB equation with quadratic running costs). *If the diffusion coefficient $\widetilde{\sigma}$ does not depend on the control, the control enters additively in the drift as in (49), and the running costs take the form*

$$\widetilde{h}(x, s, u(x, s)) = h(x, s) + \frac{1}{2}\|u(x, s)\|^2, \tag{56}$$

*then the general HJB equation in (53) can be stated in closed form as the HJB equation in (47).*

*Proof.* We formally compute

$$\inf_{\alpha \in \mathbb{R}^d} \left\{ \widetilde{h}(\cdot, \cdot, \alpha) + \widetilde{\mu}(\cdot, \cdot, \alpha) \cdot \nabla V \right\} = h + \mu \cdot \nabla V + \inf_{\alpha \in \mathbb{R}^d} \left\{ \frac{1}{2}\|\alpha\|^2 + \sigma\alpha \cdot \nabla V \right\}, \tag{57}$$

and realize that the infimum is attained when choosing $\alpha^* = -\sigma^\top \nabla V(x, t)$ for each corresponding $(x, t) \in \mathbb{R}^d \times [0, T]$, resulting in the optimal control $u^* = -\sigma^\top \nabla V$. Plugging this into the general HJB equation (53), we readily get the PDE in (47). $\qquad\square$

### A.7 Verification theorem

The *verification theorem* is a classical result in optimal control, and the proof can, for instance, be found in Nüsken & Richter (2021, Theorem 2.2), Fleming & Soner (2006, Theorem IV.4.4), and Pham (2009, Theorem 3.5.2), see also Section A.6. For the interested reader, we provide the theorem and a self-contained proof using Itô's lemma in the following. Theorem 2.4 follows with the choices $t := 0$, $b := -\mu$, $h := \mathrm{div}(\mu)$, and $g := -\log p_{X_0}$.

**Theorem A.5** (Verification theorem). *Let $V$ be a solution to the HJB equation in (47) with terminal condition $V(\cdot, T) = g$. Further, let $t \in [0, T]$ and define the set of admissible controls by*

$$\mathcal{U} := \left\{ u \in C^1(\mathbb{R}^d \times [t, T], \mathbb{R}^d) : \sup_{(x,s) \in \mathbb{R}^d \times [t,T]} \frac{\|u(x, s)\|}{1 + \|x\|} < \infty \right\}. \tag{58}$$

*For every control $u \in \mathcal{U}$ let $Z^u = (Z_s^u)_{s \in [t, T]}$ be the solution to the controlled SDE*

$$\mathrm{d}Z_s^u = (\sigma u + b)(Z_s^u, s)\,\mathrm{d}s + \sigma(s)\,\mathrm{d}B_s \tag{59}$$

*and let the cost of the control $u$ be defined by*

$$J(u; t) := \mathbb{E}\left[\int_t^T \left(h + \frac{1}{2}\|u\|^2\right)(Z_s^u, s)\,\mathrm{d}s + g(Z_T^u)\,\bigg|\,Z_t^u\right]. \tag{60}$$

*Then for every $u \in \mathcal{U}$ it holds almost surely that*

$$V(Z_t^u, t) + \mathbb{E}\left[\frac{1}{2}\int_t^T \left\|\sigma^\top \nabla V + u\right\|^2 (Z_s^u, s)\,\mathrm{d}s\,\bigg|\,Z_t^u\right] = J(u; t). \tag{61}$$

*In particular, this implies that $V(Z_t^u, t) = \min_{u \in \mathcal{U}} J(u; t)$ almost surely, where the unique minimum is attained by $u^* := -\sigma^\top \nabla V$.*

*Proof.* Let us derive the verification theorem directly from Itô's lemma, which, under suitable assumptions, states that

$$V(Z_T^u, T) - V(Z_t^u, t) = \int_t^T \left( \partial_s V + (\sigma u + b) \cdot \nabla V + \mathrm{Tr}\left(D \nabla_x^2 V\right) \right) (Z_s^u, s) \,\mathrm{d}s + S \tag{62}$$

almost surely, where

$$S := \int_t^T (\sigma^\top \nabla V)(Z_s^u, s) \cdot \mathrm{d}B_s, \tag{63}$$

see, e.g., Theorem 8.3 in Baldi (2017). Combining this with the fact that $V$ solves the HJB equation in (47) and the simple calculation

$$\frac{1}{2}\|\sigma^\top \nabla V + u\|^2 = \frac{1}{2}\left( \sigma^\top \nabla V + u \right) \cdot \left( \sigma^\top \nabla V + u \right) \tag{64a}$$

$$= \frac{1}{2}\|\sigma^\top \nabla V\|^2 + \left( \sigma^\top \nabla V \right) \cdot u + \frac{1}{2}\|u\|^2, \tag{64b}$$

shows that

$$V(Z_t^u, t) = \int_t^T \left( h + \frac{1}{2}\|u\|^2 - \frac{1}{2}\left\| \sigma^\top \nabla V + u \right\|^2 \right) (Z_s^u, s) \,\mathrm{d}s + g(Z_T^u) - S \tag{65}$$

almost surely. Under mild regularity assumptions, the stochastic integral $S$ has zero expectation conditioned on $Z_t^u$, which proves the claim. $\qquad\square$

*Remark* A.6 (Variational gap). We can interpret the term

$$\mathbb{E}\left[ \frac{1}{2} \int_t^T \left\| \sigma^\top \nabla V + u \right\|^2 (Z_s^u, s) \,\mathrm{d}s \middle| Z_t^u \right] \tag{66}$$

in (61) as the variational gap specifying the misfit of the current and the optimal control objective. In the setting of Corollary 2.5 it takes the form

$$\mathbb{E}\left[ \frac{1}{2} \int_0^T \left\| u - \sigma^\top \nabla \log \breve{p}_X \right\|^2 (Y_s^u, s) \,\mathrm{d}s \middle| Y_0 \right] \tag{67}$$

and can be compared to Huang et al. (2021, Theorem 4), where, however, the factor $1/2$ seems to be missing.

### A.8 Measures on path space

In this section, we elaborate on the path space measure perspective on diffusion-based generative modeling, as introduced in Section 2.3. Recalling our setting in Section A.1, we first state the Girsanov theorem that will be helpful in the following, see, for instance, Üstünel & Zakai (2013, Proposition 2.2.1 and Theorem 2.1.1) for a proof.

**Theorem A.7** (Girsanov theorem). *For every control $u \in \mathcal{U}$ let $Y^u = (Y_s^u)_{s \in [0,T]}$ be the solution to the controlled SDE*

$$\mathrm{d}Y_s^u = (\sigma u - \mu)(Y_s^u, s) \,\mathrm{d}s + \sigma(s) \,\mathrm{d}B_s. \tag{68}$$

*For $u, v \in \mathcal{U}$, it then holds that*

$$\log \frac{\mathrm{d}\mathbb{P}_{Y^u}}{\mathrm{d}\mathbb{P}_{Y^v}}(Y^u) = \mathcal{R}_0^{u-v}(Y^u) + \int_0^T (u - v)(Y_s^u, s) \cdot \mathrm{d}B_s \tag{69}$$

*and, in particular, that*

$$D_{\mathrm{KL}}(\mathbb{P}_{Y^u} | \mathbb{P}_{Y^v}) = \mathbb{E}\left[ \log \frac{\mathrm{d}\mathbb{P}_{Y^u}}{\mathrm{d}\mathbb{P}_{Y^v}}(Y^u) \right] = \mathbb{E}\left[ \mathcal{R}_0^{u-v}(Y^u) \right]. \tag{70}$$

Note that the expression for the KL divergence in (70) follows from the fact that, under mild regularity assumptions, the stochastic integral in (69) is a martingale and has vanishing expectation. Now, we present a lemma that specifies how the KL divergence behaves when we change the initial value of a path space measure.

**Lemma A.8** (KL divergence and disintegration). *Let $X, Y, Z$ be diffusion processes. Further, let $Z$ be such that $Z_0 \sim Y_0$ and[11] $\mathrm{d}Z = \mathrm{d}X$. Then it holds that*

$$D_{\mathrm{KL}}(\mathbb{P}_Y | \mathbb{P}_X) = D_{\mathrm{KL}}(\mathbb{P}_Y | \mathbb{P}_Z) + D_{\mathrm{KL}}(\mathbb{P}_{Y_0} | \mathbb{P}_{X_0}). \tag{71}$$

*Proof.* Let $\mathbb{P}_{Y^x}$ be the path space measure of the process $Y$ with initial condition $Y_0 = x \in \mathbb{R}^d$, let $\mathbb{P}_{Y_0}$ be the marginal at time $t = 0$, and similarly define the corresponding quantities for the processes $X$ and $Z$. Since our assumptions guarantee that $\mathbb{P}_{X^x} = \mathbb{P}_{Z^x}$ and $\mathbb{P}_{Z_0} = \mathbb{P}_{Y_0}$, the disintegration theorem, see, e.g., Léonard (2014), shows that

$$\frac{\mathrm{d}\mathbb{P}_Y}{\mathrm{d}\mathbb{P}_X} = \frac{\mathrm{d}\mathbb{P}_{Y^x}}{\mathrm{d}\mathbb{P}_{X^x}} \frac{\mathrm{d}\mathbb{P}_{Y_0}}{\mathrm{d}\mathbb{P}_{X_0}} = \frac{\mathrm{d}\mathbb{P}_{Y^x}}{\mathrm{d}\mathbb{P}_{Z^x}} \frac{\mathrm{d}\mathbb{P}_{Y_0}}{\mathrm{d}\mathbb{P}_{Z_0}} \frac{\mathrm{d}\mathbb{P}_{Y_0}}{\mathrm{d}\mathbb{P}_{X_0}} = \frac{\mathrm{d}\mathbb{P}_Y}{\mathrm{d}\mathbb{P}_Z} \frac{\mathrm{d}\mathbb{P}_{Y_0}}{\mathrm{d}\mathbb{P}_{X_0}}, \tag{72}$$

which implies the claim. $\square$

Using the previous two results, we can now provide a path space perspective on the optimal control problem in Theorem 2.4. Note that Proposition 2.6 follows directly from this result. For further connections between optimal control and path space measures, we refer to Hartmann et al. (2017); Thijssen & Kappen (2015).

**Proposition A.9** (Optimal path space measure). *We define the work functional $\mathcal{W} \colon C([0, T], \mathbb{R}^d) \to \mathbb{R}$ for all suitable stochastic processes $Y$ by*

$$\mathcal{W}(Y) := \mathcal{R}_\mu^0(Y) - \log \frac{p_{X_0}(Y_T)}{p_{X_T}(Y_0)}, \tag{73}$$

*where $\mathcal{R}_\mu^0(Y^0)$ is as in (9) with $u = 0$. Further, let the path space measure $\mathbb{Q}$ be defined via the Radon-Nikodym derivative*

$$\frac{\mathrm{d}\mathbb{Q}}{\mathrm{d}\mathbb{P}_{Y^0}} = \exp(-\mathcal{W}). \tag{74}$$

*Then it holds that $\mathbb{Q} = \mathbb{P}_{Y^{u^*}}$, where $u^* := \sigma^\top \nabla \log \overleftarrow{p}_X$ as in Corollary 2.5, and for every $u \in \mathcal{U}$ we have that*

$$\log \frac{\mathrm{d}\mathbb{P}_{Y^u}}{\mathrm{d}\mathbb{P}_{Y^{u^*}}}(Y^u) = \mathcal{R}_\mu^u(Y^u) + \int_0^T u(Y_s^u) \cdot \mathrm{d}B_s + \log \frac{p_{X_T}(Y_0^u)}{p_{X_0}(Y_T^u)}. \tag{75}$$

*In particular, the expected variational gap*

$$G(u) := \mathbb{E}\left[\log p_{X_T}(Y_0^u)\right] - \mathbb{E}\left[\log p_{X_0}(Y_T^u) - \mathcal{R}_\mu^u(Y^u)\right] \tag{76}$$

*of the ELBO in Corollary 2.5 satisfies that*

$$G(u) = D_{\mathrm{KL}}(\mathbb{P}_{Y^u} | \mathbb{P}_{Y^{u^*}}) = D_{\mathrm{KL}}(\mathbb{P}_{Y^u} | \mathbb{P}_{\overleftarrow{X}}) - D_{\mathrm{KL}}(\mathbb{P}_{Y_0^u} | \mathbb{P}_{X_T}). \tag{77}$$

*Proof.* Similar to computations in Nüsken & Richter (2021), we may compute

$$\log \frac{\mathrm{d}\mathbb{P}_{Y^u}}{\mathrm{d}\mathbb{Q}}(Y^u) = \log\left(\frac{\mathrm{d}\mathbb{P}_{Y^u}}{\mathrm{d}\mathbb{P}_{Y^0}} \frac{\mathrm{d}\mathbb{P}_{Y^0}}{\mathrm{d}\mathbb{Q}}\right)(Y^u) \tag{78a}$$

$$= \mathcal{R}_0^u(Y^u) + \int_0^T u(Y_s^u) \cdot \mathrm{d}B_s + \mathcal{R}_\mu^0(Y^u) - \log \frac{p_{X_0}(Y_T^u)}{p_{X_T}(Y_0^u)} \tag{78b}$$

$$= \mathcal{R}_\mu^u(Y^u) + \int_0^T u(Y_s^u) \cdot \mathrm{d}B_s + \log \frac{p_{X_T}(Y_0^u)}{p_{X_0}(Y_T^u)}, \tag{78c}$$

---

[11]This means that $Z$ and $X$ are governed by the same SDE, however, their initial conditions could be different.

where we used the Girsanov theorem (Theorem A.7) and (74) in (78b). This implies that

$$D_{\mathrm{KL}}(\mathbb{P}_{Y^u}|\mathbb{Q}) := \mathbb{E}\left[\log\frac{\mathrm{d}\mathbb{P}_{Y^u}}{\mathrm{d}\mathbb{Q}}(Y^u)\right] = \mathbb{E}\left[\mathcal{R}_\mu^u(Y^u) - \log p_{X_0}(Y_T^u)\right] + \mathbb{E}\left[\log p_{X_T}(Y_0^u)\right]. \tag{79}$$

Comparing to Theorem 2.4, we realize that the above KL divergence is equivalent to the control costs up to $\mathbb{E}[-V(Y_0^u, 0)] = \mathbb{E}[\log p_{X_T}(Y_0^u)]$. Using (8), we thus conclude that

$$D_{\mathrm{KL}}(\mathbb{P}_{Y^{u^*}}|\mathbb{Q}) = 0 \tag{80}$$

for $u^* = -\sigma^\top\nabla V = \sigma^\top\nabla\log\breve{p}_X$, which implies that $\mathbb{Q} = \mathbb{P}_{Y^{u^*}}$. Together with (78), this proves the claim in (75).

Now, we can express the expected variational gap $G(u)$ in (76) in terms of KL divergences. We can prove the first equality in (77) by combining the identity $\mathbb{Q} = \mathbb{P}_{Y^{u^*}}$ with (79). The second equality follows from Lemma A.8 and the observations that $Y_0^{u^*} \sim Y_0^u$ and $\mathrm{d}Y^{u^*} = \mathrm{d}\breve{X}$, see Theorem A.1. This concludes the proof. $\qquad\square$

Note that the Girsanov theorem (Theorem A.7) shows that the expected variational gap $G(u) = D_{\mathrm{KL}}(\mathbb{P}_{Y^u}|\mathbb{P}_{Y^{u^*}})$ as in (77) equals the quantity derived in Remark A.6. The following proposition states the path space measure perspective on diffusion-based generative modeling.

**Proposition A.10** (Forward KL divergence). *Let us consider the inference SDE and the controlled generative SDE as in* (14), *i.e.,*

$$\mathrm{d}Y_s = f(Y_s, s)\,\mathrm{d}s + \sigma(s)\,\mathrm{d}B_s, \quad Y_0 \sim \mathcal{D}, \quad \mathrm{d}X_s^u = \left(\breve{\sigma}\bar{u} - \breve{f}\right)(X_s^u, s)\,\mathrm{d}s + \breve{\sigma}(s)\,\mathrm{d}B_s, \tag{81}$$

*and the associated path space measures $\mathbb{P}_Y$ and $\mathbb{P}_{\breve{X}^u}$. Then the expected variational gap is given by*[12]

$$G(u) := \mathbb{E}\left[\log p_{X_T^u}(Y_0)\right] - \mathbb{E}\left[\log p_{X_0^u}(Y_T) - \mathcal{R}_{\sigma u - f}^u(Y)\right] \tag{82a}$$

$$= D_{\mathrm{KL}}(\mathbb{P}_Y|\mathbb{P}_{\breve{X}^u}) - D_{\mathrm{KL}}(\mathbb{P}_{Y_0}|\mathbb{P}_{X_T^u}). \tag{82b}$$

*Proof.* Let us consider the SDE

$$\mathrm{d}Y_s^{u,\mu} = (\sigma u - \mu)(Y_s^{u,\mu}, s)\,\mathrm{d}s + \sigma(s)\,\mathrm{d}B_s, \quad Y_0^{u,\mu} \sim \mathcal{D}. \tag{83}$$

With the choice $\mu = \sigma u - f$, we then have $Y^{u,\sigma u - f} = Y$, which actually does not depend on $u$ anymore, and $\mathrm{d}Y^{u^*,\sigma u - f} = \mathrm{d}\breve{X}^u$, where $u^* = \sigma^\top\nabla\log\breve{p}_{X^u}$. Together with the fact that $Y_0 \sim Y_0^{u^*,\sigma u - f}$, Lemma A.8 implies that

$$D_{\mathrm{KL}}(\mathbb{P}_{Y^{u,\sigma u - f}}|\mathbb{P}_{Y^{u^*,\sigma u - f}}) = D_{\mathrm{KL}}(\mathbb{P}_Y|\mathbb{P}_{\breve{X}^u}) - D_{\mathrm{KL}}(\mathbb{P}_{Y_0}|\mathbb{P}_{X_T^u}) \tag{84}$$

and Proposition A.9 (with $\mu = \sigma u - f$) yields the desired expression. $\qquad\square$

In practice, the expected variational gap $G(u)$ in (82a) cannot be minimized directly since the evidence $\log p_{X_T^u}(Y_0)$ is intractable[13] and one resorts to maximizing the ELBO instead. The same problem occurs when considering divergences other than the (forward) KL divergence $D_{\mathrm{KL}}$ in (82b). The situation is different in the context of sampling from densities, where we directly minimize the (reverse) KL divergence, see Corollary 3.1, allowing us to use other divergences, such as the log-variance divergence, see Section A.14.

---

[12]In view of Lemma A.8, the variational gap can also be written as $G(u) = D_{\mathrm{KL}}(\mathbb{P}_Y|\mathbb{P}_{Z^u})$, where $\mathrm{d}Z^u = \mathrm{d}\breve{X}^u$ and $Z_0^u = Y_0$, i.e., the process $Z^u$ is governed by the same SDE as $\breve{X}^u$, however, has the initial condition of $Y$.

[13]One could, however, use the probability flow ODE, corresponding to the choice $\lambda = 1$ in Theorem A.1, which then resembles training of a normalizing flow.

### A.9  ELBO formulations

Here we provide details on the connection of the ELBO to the denoising score matching objective. Using the reparametrization in (14), i.e. taking $\mu := \sigma u - f$, Corollary 2.5 yields that

$$\log p_{X_T^u}(Y_0) \geq \mathbb{E}\left[\log p_{X_0^u}(Y_T) - \mathcal{R}_{\sigma u - f}^u(Y)\Big|Y_0\right]. \tag{85}$$

The next lemma shows that the expected negative ELBO in (85) equals the denoising score matching objective (Vincent, 2011) up to a constant, which does not depend on the control $u$.

**Lemma A.11** (Connection to denoising score matching). *Let us define the denoising score matching objective by*

$$\mathcal{L}_{\mathrm{DSM}}(u) := \frac{T}{2}\mathbb{E}\left[\left\|u(Y_\tau, \tau) - \sigma^\top(\tau)\nabla \log p_{Y_\tau|Y_0}(Y_\tau|Y_0)\right\|^2\right]. \tag{86}$$

*Then it holds that*

$$\mathcal{L}_{\mathrm{DSM}}(u) = -\mathbb{E}\left[\log p_{X_0^u}(Y_T) - \mathcal{R}_{\sigma u - f}^u(Y)\right] + C, \tag{87}$$

*with*

$$C := \mathbb{E}\left[\log p_{X_0^u}(Y_T) + T\operatorname{div}(f)(Y_\tau, \tau) + \frac{T}{2}\left\|\sigma^\top(\tau)\nabla \log p_{Y_\tau|Y_0}(Y_\tau|Y_0)\right\|^2\right], \tag{88}$$

*where $\tau \sim \mathcal{U}([0,T])$ and $p_{Y_\tau|Y_0}$ denotes the conditional density of $Y_\tau$ given $Y_0$.*

*Proof.* The proof closely follows the one in Huang et al. (2021, Appendix A). For notational convenience, let us first define the abbreviations

$$p(x,s) := p_{Y_s|Y_0}(x|Y_0) \quad \text{and} \quad r(x,s) := u(x,s) \cdot \left(\sigma^\top(s)\nabla \log p(x,s)\right) \tag{89}$$

for every $x \in \mathbb{R}^d$ and $s \in [0,T]$. Note that we have

$$\frac{1}{2}\left\|u - \sigma^\top\nabla \log p\right\|^2 = \frac{1}{2}\|u\|^2 - r + \frac{1}{2}\left\|\sigma^\top\nabla \log p\right\|^2, \tag{90}$$

which shows that

$$\mathcal{L}_{\mathrm{DSM}}(u) = \mathbb{E}\left[T\left(\frac{1}{2}\|u\|^2 - r\right)(Y_\tau, \tau)\right] + \mathbb{E}\left[\frac{T}{2}\left\|\sigma^\top(\tau)\nabla \log p(Y_\tau, \tau)\right\|^2\right]. \tag{91}$$

Focusing on the first term on the right-hand side, Fubini's theorem and a Monte Carlo approximation establish that, under mild regularity conditions, it holds that

$$\mathbb{E}\left[T\left(\frac{1}{2}\|u\|^2 - r\right)(Y_\tau, \tau)\right] = \mathbb{E}\left[\mathcal{R}_{-f}^u(Y) + T\operatorname{div}(f)(Y_\tau, \tau)\right] - \int_0^T \mathbb{E}[r(Y_s, s)]\,\mathrm{d}s, \tag{92}$$

where the expectation on the left-hand side is over the random variable $(Y_\tau, \tau) \sim p_{Y_\tau|\tau}p_\tau$, where $p_\tau$ is the density of $\tau \sim \mathcal{U}([0,T])$ and $p_{Y_\tau|\tau}$ is the conditional density of $Y_\tau$ given $\tau$.

Together with (91), this implies that

$$\mathcal{L}_{\mathrm{DSM}}(u) = -\mathbb{E}\left[\log p_{X_0^u}(Y_T) - \mathcal{R}_{-f}^u(Y)\right] - \int_0^T \mathbb{E}[r(Y_s, s)]\,\mathrm{d}s + C. \tag{93}$$

Focusing on the term $r(Y_s, s)$ for fixed $s \in [0,T]$, it remains to show that

$$\mathbb{E}[r(Y_s, s)] = -\mathbb{E}\left[\operatorname{div}(\sigma u)(Y_s, s)\right]. \tag{94}$$

Using the identities for divergences in Section A.2, one can show that

$$\operatorname{div}(\sigma u p) - \operatorname{div}(\sigma u)p = (\sigma u) \cdot \nabla p = (\sigma u) \cdot (p\nabla \log p) = rp. \tag{95}$$

Further, Stokes' theorem guarantees that under suitable assumptions it holds that

$$\int_{\mathbb{R}^d} \mathrm{div}(\sigma u p)(x, s)\, \mathrm{d}x = 0. \tag{96}$$

Thus, using (95) and (96), we have that

$$-\mathbb{E}\left[\mathrm{div}(\sigma u)(Y_s, s)|Y_0\right] = -\int_{\mathbb{R}^d} \mathrm{div}(\sigma u)(x, s)p(x, s)\, \mathrm{d}x \tag{97a}$$

$$= \int_{\mathbb{R}^d} r(x, s)p(x, s)\mathrm{d}x = \mathbb{E}\left[r(Y_s, s)\big|Y_0\right]. \tag{97b}$$

Combining this with (93) finishes the proof. $\qquad\square$

Note that one can also establish equivalences to explicit, implicit, and sliced score matching (Hyvärinen & Dayan, 2005; Song et al., 2020), see Huang et al. (2021, Appendix A). Using the interpretation of the ELBO in terms of KL divergences, see Huang et al. (2021, Theorem 5) and also Proposition A.10, one can further derive the ELBO for discrete-time diffusion models as presented in Ho et al. (2020); Kingma et al. (2021).

### A.10 Diffusion-based sampling from unnormalized densities

In the following, we formulate an optimization problem for sampling from (unnormalized) densities. The proof and statement are similar to Theorem 2.4 and Proposition A.9 with the roles of the generative and inference SDEs interchanged. Note that Corollary 3.1 is a direct consequence of this result.

**Corollary A.12** (Reverse KL divergence). *Let $X^u$ and $Y$ be defined by (17). Then, for every $u \in \mathcal{U}$ it holds that*

$$\log \frac{\mathrm{d}\mathbb{P}_{X^u}}{\mathrm{d}\mathbb{P}_{X^{u^*}}}(X^u) = \mathcal{R}_{\bar{f}}^{\bar{u}}(X^u) + \int_0^T \bar{u}(X_s^u) \cdot \mathrm{d}B_s + \log \frac{p_{Y_T}(X_0^u)}{\rho(X_T^u)} + \log \mathcal{Z}, \tag{98}$$

*where $u^* := \sigma^\top \nabla \log p_Y$. In particular, it holds that*

$$D_{\mathrm{KL}}(\mathbb{P}_{X^u}|\mathbb{P}_{\bar{Y}}) = D_{\mathrm{KL}}(\mathbb{P}_{X^u}|\mathbb{P}_{X^{u^*}}) + D_{\mathrm{KL}}(\mathbb{P}_{X_0^u}|\mathbb{P}_{Y_T}) = \mathcal{L}_{\mathrm{DIS}}(u) + \log \mathcal{Z}, \tag{99}$$

*where*

$$\mathcal{L}_{\mathrm{DIS}}(u) := \mathbb{E}\left[\mathcal{R}_{\bar{f}}^{\bar{u}}(X^u) + \log \frac{p_{X_0^u}(X_0^u)}{\rho(X_T^u)}\right]. \tag{100}$$

*This further implies that*

$$D_{\mathrm{KL}}(\mathbb{P}_{X_0^u}|\mathbb{P}_{Y_T}) - \log \mathcal{Z} = \min_{u \in \mathcal{U}} \mathcal{L}_{\mathrm{DIS}}(u), \tag{101}$$

*and, assuming that $X_0^u \sim Y_T$, the minimizing control $u^*$ guarantees that $X_T^{u^*} \sim \mathcal{D}$.*

*Proof.* Analogously to the proof of Lemma 2.3 and noting the interchanged roles of $X^u$ and $Y$, we see that $\log \bar{p}_Y$ satisfies the HJB equation

$$\partial_t p = -\,\mathrm{Tr}\left(\bar{D}\nabla^2 p\right) + \bar{f} \cdot \nabla p + \mathrm{div}(\bar{f})p, \quad p(\cdot, T) = \log p_{Y_0} = \frac{\rho}{\mathcal{Z}}. \tag{102}$$

We can now proceed analogously to the proofs of Theorem 2.4 and Proposition 2.6. Using the identity

$$D_{\mathrm{KL}}(\mathbb{P}_{X_0^u}|\mathbb{P}_{Y_T}) = \mathbb{E}\left[\log \frac{p_{X_0^u}(X_0^u)}{p_{Y_T}(X_0^u)}\right], \tag{103}$$

this proves the statements in (99) and (101). The last statement follows from Theorem A.1 and the fact that the minimizer $u^* := \sigma^\top \nabla \log p_Y$ is the scaled score function. $\qquad\square$

Table 1: We compare the error of DDS (Vargas et al., 2023a) and DIS in estimating the log-normalizing constant and the average standard deviation on two problems. We report the average error over three independent runs. For a fair comparison, we use the same prior $\mathcal{N}(0, \mathrm{I})$, batch size $m = 2048$, and number of gradient steps $K = 20000$, and a fixed number of integrator steps $N = 256$, see Table 2. For DDS, we present the best results for different cosine schedules with $\alpha_{\max} \in \{0.25, 0.5, 1, 1.5, 2, 2.5\}$ (see Vargas et al. (2023a) for the precise definition). For comparison, we present results in their training setup as well as ours, i.e., with exponentially moving average of the parameters, clipping, and learning rate scheduler. For DIS, we also present results for the log-variance divergence, see Section A.14.

| Problem | Method | $\Delta \log \mathcal{Z} \, (rw) \downarrow$ | $\Delta \mathrm{std} \downarrow$ |
|---------|--------|---------|---------|
| GMM | DDS ($\alpha = 0.5$) | 0.176 | 2.48 |
| | DDS ($\alpha = 0.5$, our setup) | 0.090 | 2.52 |
| | DIS (KL) | 0.046 | 2.83 |
| | DIS (log-variance) | **0.013** | **0.01** |
| Funnel | DDS ($\alpha_{\max} = 0.5$) | 0.121 | 7.20 |
| | DDS ($\alpha_{\max} = 1$, our setup) | 0.046 | 6.49 |
| | DIS (KL) | **0.017** | 5.50 |
| | DIS (log-variance) | 0.021 | **5.46** |

### A.10.1 DDS as a special case of DIS

In this section, we compare the *denoising diffusion sampler* (DDS) suggested in Vargas et al. (2023a) to our *time-reversed diffusion sampler* (DIS). We present a numerical comparison in Table 1 and refer to Richter & Berner (2024, Section 3) and Vargas & Nüsken (2023) for further details. In particular, these works show that DDS can be derived using a reference process with known time-reversal. In the following, we outline how DDS can be considered a special case of DIS. To this end, we note that DDS considers the process

$$\mathrm{d}X_s = \left(-\overleftarrow{\beta}(s)X_s + 2\eta^2 \overleftarrow{\beta}(s)\left(\overleftarrow{\Phi}(X_s, s) + \frac{X_s}{\eta^2}\right)\right) \mathrm{d}s + \eta\sqrt{2\overleftarrow{\beta}(s)} \, \mathrm{d}B_s \tag{104a}$$

$$= \left(\overleftarrow{\beta}(s)X_s + 2\eta^2 \overleftarrow{\beta}(s)\overleftarrow{\Phi}(X_s, s)\right) \mathrm{d}s + \eta\sqrt{2\overleftarrow{\beta}(s)} \, \mathrm{d}B_s, \tag{104b}$$

with $X_0 \sim \mathcal{N}(0, \eta^2 \mathrm{I})$, see also (136). To ease notation, let us define $\sigma(t) := \eta\sqrt{2\beta(t)}\mathrm{I}$, $f(x, t) = -\beta(t)x$, and $u(x, t) := \sigma(t)^\top \Phi(x, t)$ to get

$$\mathrm{d}X_s^u = (-\overleftarrow{f} + \overleftarrow{\sigma}\overleftarrow{u})(X_s^u, s) \, \mathrm{d}s + \overleftarrow{\sigma}(s) \, \mathrm{d}B_s \tag{105}$$

as in (17). We added the superscript $u$ to the process in order to indicate the dependence on the control $u$. For DDS, the loss

$$\mathcal{L}_{\mathrm{DDS}}(u) := \mathbb{E}\left[\int_0^T \frac{1}{2}\left\|\overleftarrow{u}(X_s^u, s) + \frac{\overleftarrow{\sigma}(s)X_s^u}{\eta^2}\right\|^2 \mathrm{d}s + \log \frac{\mathcal{N}(X_T^u; 0, \eta^2 \mathrm{I})}{\rho(X_T^u)}\right] \tag{106}$$

is considered. In order to show the mathematical equivalence of DDS and DIS for the special choice of $\sigma$ and $f$, we may apply Itô's lemma to the function $V(x, t) = \frac{1}{2}\|x\|^2$. This yields that

$$\mathbb{E}\left[\frac{1}{2}\|X_T^u\|^2 - \frac{1}{2}\|X_0^u\|^2\right] = \mathbb{E}\left[\int_0^T \left(-\overleftarrow{f} + \overleftarrow{\sigma}\overleftarrow{u}\right)(X_s^u, s) \cdot X_s^u + \eta^2 d\overleftarrow{\beta}(s) \, \mathrm{d}s\right], \tag{107}$$

where $X^u$ is defined in (105). Noting that

$$\log \mathcal{N}(x; 0, \eta^2 \mathrm{I}) = -\frac{1}{2\eta^2}\|x\|^2 - \frac{1}{2}\log\left(2\pi\eta^2\right) \tag{108}$$

and

$$\nabla \cdot \overleftarrow{f}(x, t) = \nabla \cdot (-\overleftarrow{\beta}(t)x) = -d\overleftarrow{\beta}(t), \tag{109}$$

we can rewrite (107) as

$$\mathbb{E}\left[\log\frac{\mathcal{N}(X_T^u;0,\eta^2\mathrm{I})}{\mathcal{N}(X_0^u;0,\eta^2\mathrm{I})}\right] = \mathbb{E}\left[\int_0^T\left(\frac{\overleftarrow{f}-\overleftarrow{\sigma}\bar{u}}{\eta^2}\right)(X_s^u,s)\cdot X_s^u + \mathrm{div}\left(\overleftarrow{f}\right)(X_s^u,s)\,\mathrm{d}s\right]. \tag{110}$$

Now, we observe that the special choice of $\sigma$ and $f$ implies that

$$\left(\frac{\overleftarrow{f}-\overleftarrow{\sigma}\bar{u}}{\eta^2}\right)(X_s^u,s)\cdot X_s^u = \frac{1}{2}\|\bar{u}(X_s^u,s)\|^2 - \frac{1}{2}\left\|\bar{u}(X_s^u,s)+\frac{\overleftarrow{\sigma}(s)X_s^u}{\eta^2}\right\|^2, \tag{111}$$

which, together with (110), shows that the DDS objective in (106) can be written as

$$\mathcal{L}_{\mathrm{DDS}}(u) = \mathbb{E}\left[\mathcal{R}_{\overrightarrow{f}}^{\bar{u}}(X^u)+\log\frac{\mathcal{N}(X_0^u;0,\eta^2\mathrm{I})}{\rho(X_T^u)}\right]. \tag{112}$$

Recalling that $p_{X_0^u} = \mathcal{N}(0,\eta^2\mathrm{I})$, we obtain that $\mathcal{L}_{\mathrm{DDS}}(u) = \mathcal{L}_{\mathrm{DIS}}(u)$, where the DIS objective is given as in (19).

We emphasize, however, that the derivation of the DDS objective in (106) relies on a specific choice of the drift and diffusion coefficients $f$ and $\mu$ in (106). In contrast, the DIS objective in (19) is, in principle, valid for an arbitrary diffusion process.

### A.10.2 Optimal drifts of DIS and PIS

In this section, we analyze the optimal drift for DIS and PIS in the tractable case where the target $\rho = \mathcal{N}(m,\nu^2I)$ is a multidimensional Gaussian. This reveals that the drift of DIS can exhibit preferable numerical properties in practice. We refer to Figures 14 and 15 for visualizations.

Using the VP SDE in Section A.13 with $\eta = 1$ (as in our experiments), the optimal drift for DIS is given by

$$\mathrm{drift}_{\mathrm{DIS}}(x,t) \overset{(17)}{=} \overleftarrow{\sigma}(t)\bar{u}^*(x,t)-\overleftarrow{f}(t) = \overleftarrow{\sigma}^2(t)\underbrace{\frac{e^{-\bar{\alpha}(t)}m-x}{1+e^{-2\bar{\alpha}(t)}(\nu^2-1)}}_{=\nabla\log\overleftarrow{p}_Y(x,t)}+\frac{1}{2}\overleftarrow{\sigma}^2(t)x, \tag{113}$$

where $\alpha$ is as in (138), see (137). Note that for $\nu = 1$, we can rewrite the score as

$$\nabla\log\overleftarrow{p}_Y(x,t) = e^{-\bar{\alpha}(t)}m-x = (1-e^{-\bar{\alpha}(t)})\nabla\log p_{X_0^u}(x)+e^{-\bar{\alpha}(t)}\nabla\log\rho(x), \tag{114}$$

which is reminiscent of our initialization given by a linear interpolation and, in fact, the same for the (admittedly exotic) choice $\sigma^2(t) = \frac{2}{T-t}$, see Section A.13.

In comparison, the optimal drift of PIS (using $f = 0$, i.e., a pinned Brownian motion scaled by $\sigma \in (0,\infty)$ as in the original paper) can be calculated via the Feynman-Kac formula (Zhang & Chen, 2022a; Tzen & Raginsky, 2019) and is given by

$$\mathrm{drift}_{\mathrm{PIS}}(x,t) = \sigma\bar{u}^*(x,t) = \frac{\sigma^2T}{t\nu^2+\sigma^2T(T-t)}\left(m-\frac{Tx}{t}\right)+\frac{x}{t}, \tag{115}$$

see Vargas et al. (2023a, Appendix A.2). Due to the initial delta distribution, the drift of PIS can become unbounded for $t \to 0$, which causes instabilities in practice. Note that this is not the case for DIS.

### A.11 PDE-based methods for sampling from unnormalized densities

In principle, the different perspectives on diffusion-based generative modeling introduced in Section 2 allow for different numerical methods. While our suggested method for sampling from (unnormalized) densities in Section 3 directly follows from the optimal control viewpoint, an alternative route is motivated by the PDE perspective outlined in Section 2.1. While classical numerical methods for approximating PDEs suffer from

the curse of dimensionality, we shall briefly discuss methods that may be computationally feasible even in higher dimensions.

Let us first rewrite the PDEs from Section 2.1 in a way that makes their numerical approximation feasible. To this end, we recall that the Fokker-Planck equation is a linear PDE and its time-reversal can be written as a generalized Kolmogorov backward equation. Specifically, in the setting of Section 3, it holds for the time-reversed density $\breve{p}_Y$ that

$$\partial_t \breve{p}_Y = \mathrm{div}\left(-\mathrm{div}\left(\breve{D}\breve{p}_Y\right) + \breve{f}\breve{p}_Y\right) = -\mathrm{Tr}\left(\breve{D}\nabla^2 \breve{p}_Y\right) + \breve{f}\cdot\nabla\breve{p}_Y + \mathrm{div}(\breve{f})\breve{p}_Y, \tag{116}$$

analogously to (5). We note that, due to the linearity of the PDE in (116), the scaled density $p := \mathcal{Z}\breve{p}_Y$ also satisfies a Kolmogorov backward equation given by

$$\partial_t p = -\mathrm{Tr}\left(\breve{D}\nabla^2 p\right) + \breve{f}\cdot\nabla p + \mathrm{div}(\breve{f})p, \quad p(\cdot, T) = \rho. \tag{117}$$

The corresponding HJB equation following from the Hopf–Cole transformation $V := -\log\left(\mathcal{Z}\breve{p}_Y\right)$ (as outlined in Section A.5) equals

$$\partial_t V = -\mathrm{Tr}(\breve{D}\nabla^2 V) + \breve{f}\cdot\nabla V - \mathrm{div}(\breve{f}) + \frac{1}{2}\left\|\breve{\sigma}^\top\nabla V\right\|^2, \quad V(\cdot, T) = -\log\rho, \tag{118}$$

in analogy to Lemma 2.3, however, with a modified terminal condition since we omitted the normalizing constant. Now, as $\rho$ is known, viable strategies to learn $u^*$ would be to solve the PDEs for $p$ or $V$ via deep learning.

The general idea for the approximation of PDE solutions via deep learning is to define loss functionals $\mathcal{L}$ such that

$$\mathcal{L}(p) \text{ is minimal } \iff p \text{ is a solution to (117)}, \tag{119}$$

or, respectively,

$$\mathcal{L}(V) \text{ is minimal } \iff V \text{ is a solution to (118)}. \tag{120}$$

This variational perspective then allows to parametrize the solution $p$ or $V$ by a neural network and to minimize suitable estimator versions of the loss functional $\mathcal{L}$ via gradient-based methods. We can then use automatic differentiation to compute

$$u^* := \sigma^\top\nabla\log p_Y = \sigma^\top\nabla\log\breve{p} = \sigma^\top\nabla\breve{V} \tag{121}$$

as the score is invariant to rescaling. This adds an alternative to directly optimizing the control costs in (19) in order to approximately learn $u \approx u^*$.

### A.11.1   Physics informed neural networks

A general approach to obtain suitable loss functionals $\mathcal{L}$ for learning the PDEs, dating back to the 1990s (see, e.g., Lagaris et al. (1998)), is often referred to as *Physics informed neural networks* (PINNs) or *deep Galerkin method* (DGM). These approaches have recently become popular for approximating solutions to PDEs via a combination of Monte Carlo simulation and deep learning (Nüsken & Richter, 2023; Raissi et al., 2017; Sirignano & Spiliopoulos, 2018). The idea is to minimize the squared residual of the PDE in (118) for an approximation $\widetilde{V} \approx V$, i.e.,

$$\mathcal{L}_{\mathrm{PDE}}(\widetilde{V}) = \mathbb{E}\left[\left(\partial_t\widetilde{V} + \mathrm{Tr}\left(\breve{D}\nabla^2\widetilde{V}\right) - \breve{f}\cdot\nabla\widetilde{V} + \mathrm{div}(\breve{f}) - \frac{1}{2}\left\|\breve{\sigma}^\top\nabla\widetilde{V}\right\|^2\right)^2(\xi, \tau)\right], \tag{122}$$

as well as the squared residual of the terminal condition, i.e.,

$$\mathcal{L}_{\mathrm{boundary}}(\widetilde{V}) = \mathbb{E}\left[\left(\widetilde{V}(\xi, T) + \log\rho(\xi)\right)^2\right], \tag{123}$$

where $(\xi, \tau)$ is a suitable random variable distributed on $\mathbb{R}^d \times [0, T]$. This results in a loss

$$\mathcal{L}_{\mathrm{PINN}}(\widetilde{V}) = \mathcal{L}_{\mathrm{PDE}}(\widetilde{V}) + c\,\mathcal{L}_{\mathrm{boundary}}(\widetilde{V}), \tag{124}$$

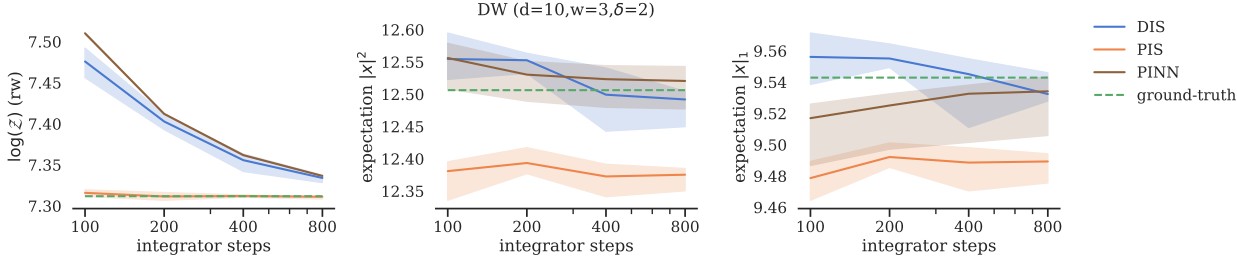

Figure 7: We use the settings in Figures 4 and 5 and display the estimates (median and interquartile range over 10 training seeds) of DIS, PIS, and the PINN method for $\log \mathcal{Z}$ and the expected values of two choices of $\gamma$ in (148). We find that the PINN method can provide competitive results.

where $c \in (0, \infty)$ is a suitably chosen penalty parameter. Using automatic differentiation, one can then compute an approximation of the control as in (121). In Figure 7, we compare this approach to DIS and PIS on a 10-dimensional double well example. While DIS provides better results overall, the PINN approach outperforms PIS and holds promise for further development. A clear advantage of PINN is the fact that no time-discretization is necessary. On the other hand, one needs to compute higher-order derivatives in the PINN objective (122) during training, and one also needs to evaluate the derivative of $\widetilde{V}$ to obtain the approximated score during sampling.

### A.11.2 BSDE-based methods and Feynman-Kac formula

To reduce the computational cost for the higher-order derivatives in the PINN objective, one can also develop loss functionals $\mathcal{L}$ for our specific PDEs by considering suitable stochastic representations. This can be done by applying Itô's lemma to the corresponding solutions. For instance, for the HJB equation in (118), we obtain

$$-\log \rho(X_T) = V(\xi, \tau) + \mathcal{R}_{-\bar{f}}^{\bar{\sigma}^\top \nabla V}(X) + \mathcal{S}_V(X), \tag{125}$$

where we abbreviate

$$\mathcal{S}_p(X) := \int_\tau^T \left( \sigma^\top \nabla p \right) (X_s, s) \cdot \mathrm{d}B_s. \tag{126}$$

In the above, the stochastic process $X$ is given by the SDE

$$\mathrm{d}X_s = -\bar{f}(X_s, s)\,\mathrm{d}s + \bar{\sigma}\mathrm{d}B_s, \quad X_\tau \sim \xi, \tag{127}$$

where $(\xi, \tau)$ is again a suitable random variable distributed on $\mathbb{R}^d \times [0, T]$. Minimizing the squared residual in (125) for an approximation $\widetilde{V} \approx V$, we can thus define a viable loss functional by

$$\mathcal{L}_{\mathrm{BSDE}}(\widetilde{V}) := \mathbb{E}\left[ \left( \widetilde{V}(\xi, \tau) + \mathcal{R}_{-\bar{f}}^{\bar{\sigma}^\top \nabla \widetilde{V}}(X) + \mathcal{S}_{\widetilde{V}}(X) + \log \rho(X_T) \right)^2 \right], \tag{128}$$

see, e.g., Richter (2021); Nüsken & Richter (2021). The name of the loss functional $\mathcal{L}_{\mathrm{BSDE}}$ stems from the fact that (129) can also be seen as a backward stochastic differential equation (BSDE). A method to combine the PINN loss in A.11.1 with the BSDE-based loss in (128) is presented in Nüsken & Richter (2023), leading to the so-called *diffusion loss*, which allows to consider arbitrary SDE trajectory lengths.

One can argue in a similar fashion for the linear PDE in (117), where Itô's lemma establishes that

$$\rho(X_T) = p(\xi, \tau) + \int_\tau^T \left( \mathrm{div}(\bar{f})p \right) (X_s, s)\,\mathrm{d}s + \mathcal{S}_p(X). \tag{129}$$

However, for the linear Kolmogorov backward equation, we can also make use of the celebrated Feynman-Kac formula, i.e.,

$$p(\xi, \tau) = \mathbb{E}\left[ \exp\left( -\int_\tau^T \mathrm{div}(\bar{f})(X_s, s)\,\mathrm{d}s \right) \rho(X_T) \bigg| (\xi, \tau) \right] \tag{130}$$

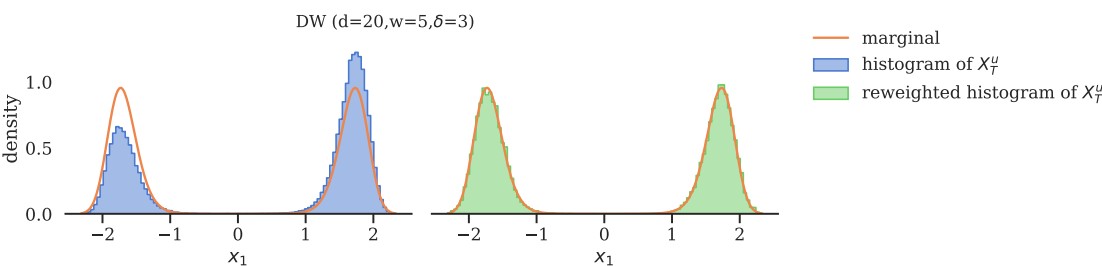

Figure 8: Illustration of importance sampling in path space. The left panel shows a histogram of the first component of the controlled process $X_T^u$ at terminal time $T$ compared to the corresponding marginal of the target density $\rho/\mathcal{Z}$. While we cannot correct the error caused by $p_{X_0^u} \approx p_{Y_T}$, we can mitigate the error in the control $u \approx u^*$ by reweighting $X_T^u$ with the importance weights $\widetilde{w}^u$, see Section A.12. The right panel shows that this indeed results in an improved approximation of the target density.

to establish the loss

$$\mathcal{L}_{\mathrm{FK}}(\widetilde{p}) = \mathbb{E}\left[\left(\left(\widetilde{p}(\xi,\tau) - \exp\left(-\int_\tau^T \operatorname{div}(\overleftarrow{f})(X_s,s)\,\mathrm{d}s\right)\right)\rho(X_T)\right)^2\right],\tag{131}$$

see, e.g., Berner et al. (2020); Richter & Berner (2022); Beck et al. (2021). Since the stochastic integral $\mathcal{S}_p(X)$ has vanishing expectation conditioned on $(\xi,\tau)$, it can also be included in the FK-based loss in (131) to reduce the variance of corresponding estimators and improve the performance, see Richter & Berner (2022).

### A.12 Importance sampling in path space

In practice, we will usually not have $u^*$ available but must rely on an approximation $u \approx u^*$. In consequence, this then yields samples $X_T^u$ that are only approximately distributed according to the target density $\rho$ and thus leads to biased estimates. However, we can correct for this bias by employing importance sampling.

Importance sampling is a classical method for variance-reduction in Monte Carlo approximation that leads to unbiased estimates of corresponding quantities of interest (Liu & Liu, 2001). The idea is to sample from a proposal distribution and correct for the corresponding bias by a likelihood ratio between the proposal and the target distribution. While importance sampling is commonly used for densities, one can also employ it in path space, thereby reweighting continuous trajectories of a stochastic process (Hartmann et al., 2017). In our application, we can make use of this in order to correct for the bias introduced by the fact that $u$ is only an approximation of the optimal $u^*$. Note that, in general, it holds for any suitable functional $\varphi : C([0,T],\mathbb{R}^d) \to \mathbb{R}$ that

$$\mathbb{E}\left[\varphi(X^{u^*})\right] = \mathbb{E}\left[\varphi(X^u)\frac{\mathrm{d}\mathbb{P}_{X^{u^*}}}{\mathrm{d}\mathbb{P}_{X^u}}(X^u)\right],\tag{132}$$

i.e., the expectation can be computed w.r.t. samples from an optimally controlled process $X^{u^*}$, even though the actual samples originate from the process $X^u$. The weights necessary for this can be calculated via

$$w^u := \frac{\mathrm{d}\mathbb{P}_{X^{u^*}}}{\mathrm{d}\mathbb{P}_{X^u}}(X^u) = \frac{1}{\mathcal{Z}}\exp\left(-\mathcal{R}_{\overleftarrow{f}}^{\widetilde{u}}(X^u) - \int_0^T \widetilde{u}(X_s^u)\cdot\mathrm{d}B_s - \log\frac{p_{Y_T}(X_0^u)}{\rho(X_T^u)}\right),\tag{133}$$

where we used the Radon-Nikodym derivative from Proposition A.9. The identity in (132) then yields an unbiased estimator, however, with potentially increased variance. In fact, the variance of importance sampling estimators can scale exponentially in the dimension $d$ and in the deviation of $u$ from the optimal $u^*$, thereby being very sensitive to the approximation quality of the control $u$, see, for instance, Hartmann & Richter (2021).

Note that, in practice, we do not know $\mathcal{Z}$, i.e., we need to consider the unnormalized weights

$$\widetilde{w}^u := w^u\mathcal{Z}.\tag{134}$$

We can then make use of the identity

$$\mathbb{E}\left[\varphi(X^{u^*})\right] = \frac{\mathbb{E}\left[\varphi(X^u)\widetilde{w}(X^u)\right]}{\mathbb{E}\left[\widetilde{w}(X^u)\right]}. \tag{135}$$

While for normalized weights as in (133) the variance of the corresponding importance sampling estimator vanishes at the optimum $u = u^*$, this is, in general, not the case for the unnormalized weights defined in (134).

In implementations, we introduce a bias in the importance sampling estimator by simulating the SDE $X^u$ using a time-discretization, such as the Euler-Maruyama scheme, see Section A.13. Moreover, the quantity $p_{Y_T}$ in (133) is typically intractable and we need to use the approximation $p_{X_0^u} \approx p_{Y_T}$ as in the loss $\mathcal{L}_{\mathrm{DIS}}$, see Corollary 3.1. We have illustrated the effect of importance sampling in Figure 8.

### A.13 Details on implementation

In this section, we provide further details on the implementation.

**SDE:** For the inference SDE $Y$ we employ the variance-preserving (VP) SDE from Song et al. (2021) given by

$$\sigma(t) := \eta\sqrt{2\beta(t)}\,\mathrm{I} \quad \text{and} \quad f(x,t) := -\beta(t)x \tag{136}$$

with $\beta \in C([0,T],(0,\infty))$ and $\eta \in (0,\infty)$. Note that this constitutes an Ornstein-Uhlenbeck process with conditional density

$$p_{Y_t|Y_0}(\cdot|Y_0) = \mathcal{N}\left(e^{-\alpha(t)}Y_0, \eta^2\left(1 - e^{-2\alpha(t)}\right)\mathrm{I}\right), \tag{137}$$

where

$$\alpha(t) := \int_0^t \beta(s)\mathrm{d}s. \tag{138}$$

In particular, we observe that

$$p_{Y_T} = \mathbb{E}\left[p_{Y_T|Y_0}(\cdot|Y_0)\right] \approx \mathcal{N}\left(0, \eta^2\mathrm{I}\right) \tag{139}$$

for suitable $\beta$ and sufficiently large $T$. In practice, we choose the schedule

$$\beta(t) := \frac{1}{2}\left(\left(1 - \frac{t}{T}\right)\sigma_{\min} + \frac{t}{T}\sigma_{\max}\right) \tag{140}$$

with $\eta = 1$ and sufficiently large $0 < \sigma_{\min} < \sigma_{\max}$. This motivates our choice $X_0^u \sim \mathcal{N}(0,\mathrm{I})$. We can numerically check whether (the unconditional) $Y_T$ is indeed close to a standard normal distribution. To this end, we consider samples from $\mathcal{D}$ and let the process $Y$ run according to the inference SDE in (3). We can now compare the empirical distributions of the different components, which each need to follow a one-dimensional Gaussian. Figure 9 shows that this is indeed the case.

**Model:** Similar to PIS, we employ the initial score of the inference SDE, i.e., the score of the data distribution, $\nabla \log p_{Y_0} = \nabla \log \rho$, in the parametrization of our control. Additionally, in our DIS method, we also make use of the (tractable) initial score of the generative SDE $\nabla \log p_{X_0^u} = \nabla \log \mathcal{N}(0,\mathrm{I})$. Specifically, we choose

$$u_\theta(x,t) := \Phi_\theta^{(1)}(x,t) + \Phi_\theta^{(2)}(t)\sigma(t)s(x,t), \tag{141}$$

where

$$s(x,t) := \frac{t}{T}\nabla \log p_{X_0^u}(x) + \left(1 - \frac{t}{T}\right)\nabla \log \rho(x) \tag{142}$$

and $\Phi_\theta^{(1)}$ and $\Phi_\theta^{(2)}$ are neural networks with parameters $\theta$. We use the same network architectures as PIS, which in particular uses a Fourier feature embedding (Tancik et al., 2020) for the time variable $t$. We initialize $\theta^{(0)}$ such that $\Phi_{\theta^{(0)}}^{(1)} \equiv 0$ and $\Phi_{\theta^{(0)}}^{(2)} \equiv 1$. The parametrization in (141) establishes that our initial control (approximately) matches the optimal control at the initial and terminal times, i.e.,

$$u_{\theta^{(0)}}(\cdot,0) = \sigma(0)^\top\nabla \log \rho = \sigma(0)^\top\nabla \log p_{Y_0} = u^*(\cdot,0) \tag{143}$$

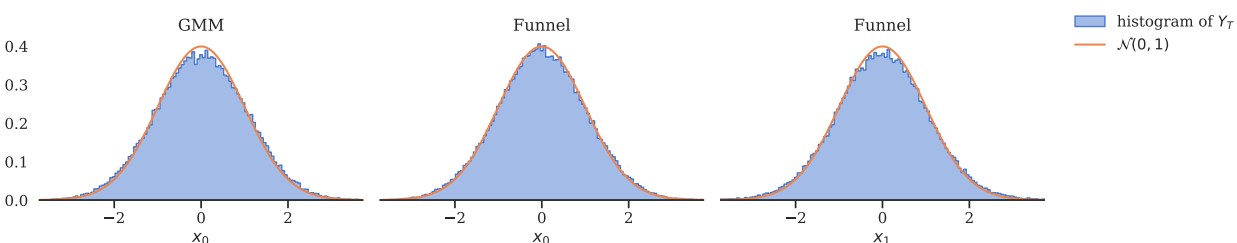

Figure 9: Histogram of the first component of $Y_T$ for the Gaussian mixture model and the first two coordinates of the Funnel distribution (in blue) when starting the corresponding process $Y$ in $Y_0 \sim \mathcal{D}$ and using $N = 800$ steps for the simulation, compared to the density of a standard normal (in orange). We do not depict the double well as we cannot sample from the ground truth distribution.

and

$$u_{\theta^{(0)}}(\cdot, T) = \sigma(T)^\top \nabla \log p_{X_0^u} \approx \sigma(T)^\top \nabla \log p_{Y_T} = u^*(\cdot, T). \tag{144}$$

In our experiments, we found it beneficial to detach the evaluation of $s$ from the computational graph and to use a clipping schedule for the output of the neural networks.

**Training:** We minimize the mapping $\theta \mapsto \widehat{\mathcal{L}}_{\mathrm{DIS}}(u_\theta)$ using a variant of gradient descent, i.e., the Adam optimizer (Kingma & Ba, 2014), where $\widehat{\mathcal{L}}_{\mathrm{DIS}}$ is a Monte Carlo estimator of $\mathcal{L}_{\mathrm{DIS}}$ in (19). Given that we are now optimizing a reverse KL divergence, the expectation in $\mathcal{L}_{\mathrm{DIS}}(u_\theta)$ is over the controlled process $X^{u_\theta}$ in (17) and we need to discretize $X^{u_\theta}$ and the running costs $\mathcal{R}_{\bar{f}}^{\bar{u}_\theta}(X^{u_\theta})$ on a time grid $0 = t_0 < \cdots < t_N = T$. We use the *Euler-Maruyama* (EM) scheme given by

$$\widehat{X}_{n+1}^\theta = \widehat{X}_n^\theta + \left(\bar{\sigma}\bar{u}_\theta - \bar{f}\right)(\widehat{X}_n^\theta, t_n)\Delta t + \bar{\sigma}(t_n)\underbrace{(B_{t_{n+1}} - B_{t_n})}_{\sim \mathcal{N}(0, \Delta t \mathrm{I})}, \tag{145}$$

where $\Delta t \coloneqq T/N = t_{n+1} - t_n$ is the step size. Given $\widehat{X}_0^\theta \sim X_0^{u_\theta}$, it can be shown that $\widehat{X}_N^\theta$ convergences to $X_T^{u_\theta}$ for $\Delta t \to 0$ in an appropriate sense (Kloeden & Platen, 1992). This motivates why we increase the number of steps $N$ for our methods when the training progresses, see Table 2. A smaller step size typically leads to a better approximation but incurs increased computational costs. We also trained separate models with a fixed number of steps, but did not observe benefits justifying the additional computational effort, see Figure 10.

We compute the derivatives w.r.t. $\theta$ by automatic differentiation. In a memory-restricted setting, one could alternatively use the *stochastic adjoint sensitivity method* (Li et al., 2020; Kidger et al., 2021) to compute the gradients using adaptive SDE solvers. For a given time constraint, we did not observe better performance of stochastic adjoint sensitivity methods, higher-order (adaptive) SDE solvers, or the exponential integrator by Zhang & Chen (2022b, Section 5) over the Euler-Maruyama scheme. Our training scheme yields better results compared to the models obtained when using the default configuration in the code of Zhang & Chen (2022a), see Figure 11. For our comparisons, we thus trained models for PIS according to our scheme, see also Table 2. Note that in Section A.11 we present an alternative approach of training a control $u_\theta$ based on *physics-informed neural networks* (PINNs).

**Sampling:** In order to reduce the variance, we compute an exponential moving average $\bar{\theta}$ of the parameters $(\theta^{(k)})_{k=1}^K$ in Algorithm 1 during training and use it for evaluation. We evaluate our model by sampling from the prior $\widehat{X}_0^{\bar{\theta}} \sim \mathcal{N}(0, \mathrm{I})$ and simulating the generative SDE using the EM scheme, such that $\widehat{X}_N^{\bar{\theta}}$ represents an approximate sample from $\mathcal{D}$. Note that we can, in principle, choose the number of steps for the EM scheme independent from the steps used for training. We used an increasing number of steps during training, and our trained model provides good results for a range of steps, see Figures 4 and 5. Thus, we amortized the

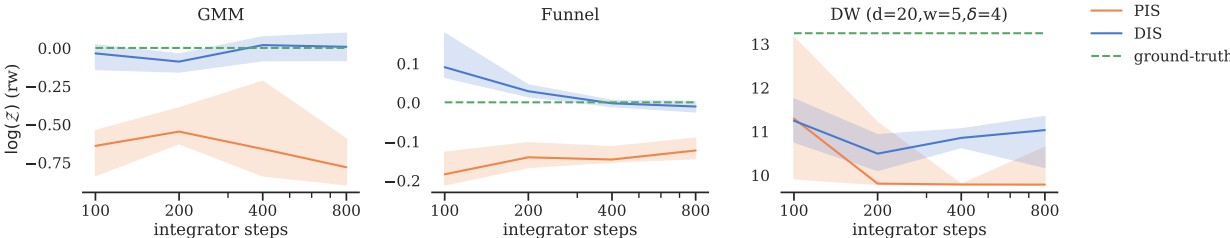

Figure 10: We use the same setting as in Figure 4, except that for each choice of SDE integrator steps $N$ a separate model is trained, i.e., all gradient steps use the same step size $\Delta t$. This yields qualitatively similar results. For our other experiments, we thus amortize the cost and train a model which can be evaluated for a range of steps $N$.

cost of training different models for different step sizes. Finally, note that similar to Song et al. (2021), one could alternatively obtain approximate samples from $\mathcal{D}$ by simulating the *probability flow ODE*, originating from the choice $\lambda = 1$ in Theorem A.1, using suitable ODE solvers (Zhang & Chen, 2022b).

**Evaluation:** We evaluate the performance of our proposed method, DIS, against PIS on the following two metrics:

- *approximating normalizing constants:* We obtain a lower bound[14] on the log-normalizing constant $\log \mathcal{Z}$ by evaluating the negative control costs, i.e.,

$$\log \mathcal{Z} \geq \log \mathcal{Z} - D_{\mathrm{KL}}(\mathbb{P}_{X_0^u} | \mathbb{P}_{Y_T}) \geq -\mathcal{L}_{\mathrm{DIS}}(u_{\bar{\theta}}) \tag{146}$$

  see (20). We note that, similar to the PIS method, adding the stochastic integral (which has zero expectation), i.e., considering

$$\mathcal{R}_{\bar{\mu}}^{\bar{u}}(X^u) + \log \frac{p_{X_0^u}(X_0^u)}{\rho(X_T^u)} + \int_0^T u(Y_s^u, s) \cdot \mathrm{d}B_s \tag{147}$$

  yields an estimator with lower variance, see also Corollary A.12. We will compare unbiased estimates using *importance sampling* in path space, see Section A.12.

- *approximating expectations and standard deviations:* In applications, one is often interested in computing expected values of the form

$$\Gamma := \mathbb{E}\left[\gamma(Y_0)\right], \tag{148}$$

  where $Y_0 \sim \mathcal{D}$ follows some distribution and $\gamma : \mathbb{R}^d \to \mathbb{R}$ specifies an observable of interest. We consider $\gamma(x) := \|x\|^2$ and $\gamma(x) := \|x\|_1 = \sum_{i=1}^d |x_i|$. We approximate (148) by creating samples $(\widehat{X}_N^{\bar{\theta},(k)})_{k=1}^K$ from our model as described in the previous paragraph, where $K \in \mathbb{N}$ specifies the sample size. We can then approximate $\Gamma$ in (148) via Monte Carlo sampling by

$$\widehat{\Gamma} := \frac{1}{K} \sum_{k=1}^K \gamma(\widehat{X}_N^{\bar{\theta},(k)}). \tag{149}$$

  Given a reference solution $\Gamma$ of (148), we can now compute the relative error $|(\widehat{\Gamma} - \Gamma)/\Gamma|$, which can be viewed as an evaluation of the sample quality on a global level. Analogously, we analyze the error when approximating coordinate-wise standard deviations of the target distribution $Y_0$ using the samples $(\widehat{X}_N^{\bar{\theta},(k)})_{k=1}^K$.

---

[14]We note that an Euler-Maruyama discretization of $-\mathcal{L}_{\mathrm{DIS}}(u_{\bar{\theta}})$ in (146) can potentially overestimate the log-normalizing constant $\log \mathcal{Z}$. We refer to Vargas & Nüsken (2023) for reformulations that guarantee a lower bound using backward integration.

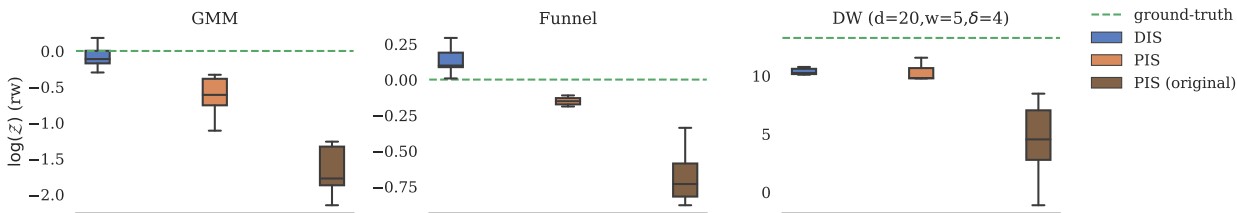

Figure 11: The boxplot compares our models to the results obtained from the repository of Zhang & Chen (2022a) using their settings. As they only train with $N = 100$ steps and present approximations of the log-normalizing constants, we show corresponding results. The figure shows that our implementation of the PIS method already provides significantly improved results. Compared to their settings, we train until convergence with larger batch sizes, use increasingly finer step sizes, a clipping schedule, and an exponential moving average of the parameters.

### A.14 Further numerical results

In this section, we present further empirical results and comparisons.

**Other divergences on path space:** Our principled framework of considering path space measures in Section 2.3 allows us to consider divergences other than the (reverse) KL divergence, which is known to suffer from mode collapse (Minka et al., 2005; Midgley et al., 2022) or a potentially high variance of Monte Carlo estimators (Roeder et al., 2017). We conducted preliminary experiments and want to highlight results using the *log-variance divergence* that has been suggested in Nüsken & Richter (2021) and often exhibits favorable numerical properties. In fact, we can see in Figure 6 that, for some experiments, this divergence can improve performance significantly – sometimes by more than an order of magnitude. We, therefore, suspect that further improvements will be possible by leveraging our path space perspective, see Richter & Berner (2024) for an extensive comparison.

**Other parametrizations and initializations:** We tried a number of parametrizations and initializations for our neural networks. Specifically, we tried to initialize the control $u$ such that the generative SDE $X^u$

1. has approximately zero drift (as in PIS),

2. corresponds to the SDE of the unadjusted Langevin algorithm (ULA),

3. has the correct drift at the initial and terminal times, i.e., the interpolation described in (141).

For Item 1, we also experimented with omitting the score $\nabla \log \rho$ in the parametrization. We present a comparison of the learned densities for the Funnel problem in Figure 12 and a GMM problem in Figure 13, where we chose a non-symmetric GMM example to emphasize the different behavior. While there are problem-specific performance differences, we observed that all the parametrizations incorporating the score $\nabla \log \rho$ yielded qualitatively similar final results. Thus, we opted for the most stable combination, i.e., Item 3, which is detailed in (141) above. As further benchmarks and metrics, we also present the results of running ULA, see the next paragraph for details, and the Sinkhorn distance[15] (Cuturi, 2013) to ground truth samples. Finally, we provide a comparison of the corresponding learned drifts in Figure 14 and Figure 15. We note that the learned drifts are closer to the optimal drift for larger $t \in [0, T]$. It would be an interesting future direction to try different loss weighting schemes (as have been employed for the denoising score matching objective) to facilitate learning the score for small $t$.

**Unadjusted Langevin Algorithm (ULA):** Note that Langevin-based algorithms (which are not model-based), such as ULA or the Metropolis-adjusted Langevin algorithm (MALA), are only guaranteed to converge

---

[15]Our implementation is based on `https://github.com/fwilliams/scalable-pytorch-sinkhorn` with the default parameters.

to the target distribution $\mathcal{D}$ in infinite time. In contrast, DIS and PIS are guaranteed to sample from the target after finite time $T$ (given that the learned model converged). This is particularly relevant for distributions $\mathcal{D}$ for which the convergence speed of the Langevin-based algorithms is slow – which often appears if the modes of the corresponding densities are very disconnected (or, equivalently, if the potential consists of multiple minima, which are separated by rather high "energy barriers"). In fact, by Kramer's law, the time to escape such a minimum via the Langevin SDE satisfies the large deviations asymptotics $\mathbb{E}[\tau] \asymp \exp(2\Delta\Psi/\eta)$, where $\tau$ is the exit time from the well, $\Delta\Psi$ is the energy barrier, i.e., the difference between the minimum and the corresponding maximum in the potential function, and $\eta$ is a temperature scaling (Berglund, 2011). Here, the crucial part is the exponential scaling in the energy barriers, which results in the Langevin diffusion "being stuck" in the potential minima for a very long time. We visualize such behavior in the experiments provided in Figures 12 and 13. In our GMM experiment, only three modes are sufficiently covered, even for large terminal times, such as $T = 2000$, which takes roughly ten times longer than DIS or PIS to sample 6000 points on a GPU. Trying larger step sizes leads to bias and divergence, as can be seen in the last plot. We observe similar results for the Funnel distribution, although we need smaller step sizes to even converge. We note that one can counteract the bias by additionally using MCMC methods (e.g., MALA), but it remains hard to tune the parameters for a given problem.

Table 2: Hyperparameters

| **DIS SDE** | |
| --- | --- |
| inference SDE (corresponding to $Y$) | Variance-Preserving SDE with linear schedule (Song et al., 2021) |
| min. diffusivity $\sigma_{\min}$ | 0.1 |
| max. diffusivity $\sigma_{\max}$ | 10 |
| terminal time $T$ | 1 |
| initial distribution $X_0^u$ | $\mathcal{N}(0, \mathrm{I})$ (truncated to 99.99% of mass) |
| **PIS SDE** (Zhang & Chen, 2022a) | |
| uncontrolled process $X^0$ | scaled Brownian motion |
| drift $\bar{f}$ | 0 (constant) |
| diffusivity $\bar{\sigma}$ | $\sqrt{0.2}$ (constant) |
| terminal time $T$ | 5 |
| initial distribution | Dirac delta $\delta_0$ at the origin |
| **SDE Solver** | |
| type | Euler-Maruyama (Kloeden & Platen, 1992) |
| steps $N$ (see also figure descriptions) | $[100, 200, 400, 800]$ (each for $1/4$ of the total gradient steps $K$) |
| **Training** | |
| optimizer | Adam (Kingma & Ba, 2014) |
| weight decay | $10^{-7}$ |
| learning rate | 0.005 |
| batch size $m$ | 2048 |
| gradient clipping | 1 ($\ell^2$-norm) |
| clip $\Phi_\theta^{(1)}, s \in [-c, c]$, $\Phi_\theta^{(2)} \in [-c, c]^d$ | $c = 10$ (step $\leq 200$), $c = 50$ ($200 < \text{step} \leq 400$), $c = 250$ (else) |
| gradient steps $K$ | 20000 ($d \leq 10$), 80000 (else) |
| framework | PyTorch (Paszke et al., 2019) |
| GPU | Tesla V100 (32 GiB) |
| number of seeds | 10 |
| **Network** fourier-emb$_c$ (Tancik et al., 2020) | |
| dimensions | $[0, T] \to \mathbb{R}^{2c} \to \mathbb{R}^c \to \mathbb{R}^c$ |
| architecture | linear $\circ [\varrho \circ \text{linear}] \circ \text{fourier-features}$ |
| activation function $\varrho$ | GELU (Hendrycks & Gimpel, 2016) |
| **Network** $\Phi_\theta^{(1)}$ (Zhang & Chen, 2022a) | |
| width $c$ | 64 ($d \leq 10$), 128 (else) |
| dimensions | $\mathbb{R}^d \times [0, T] \to \mathbb{R}^c \times \mathbb{R}^c \to \mathbb{R}^c \to \mathbb{R}^c \to \mathbb{R}^c \to \mathbb{R}^d$ |
| architecture | linear $\circ [\varrho \circ \text{linear}] \circ [\varrho \circ \text{linear}] \circ [\varrho \circ \text{sum}] \circ [\text{linear}, \text{fourier-emb}_c]$ |
| activation function | GELU (Hendrycks & Gimpel, 2016) |
| bias initialization in last layer | 0 |
| weight initialization in last layer | 0 |
| **Network** $\Phi_\theta^{(2)}$ (Zhang & Chen, 2022a) | |
| dimensions | $[0, T] \to \mathbb{R}^{64} \to \mathbb{R}^{64} \to \mathbb{R}$ |
| architecture | $[\text{linear} \circ \varrho] \circ [\text{linear} \circ \varrho] \circ \text{fourier-emb}_{64}$ |
| activation function $\varrho$ | GELU (Hendrycks & Gimpel, 2016) |
| bias initialization in last layer | 1 |
| weight initialization in last layer | 0 |
| **Evaluation** | |
| exponentially moving average $\bar{\theta}$ | last 1500 steps (updated every 5-th) with decay $1 - \frac{1}{1+\text{ema\_step}\,/\,0.9}$ |
| samples $M$ | 6000 |

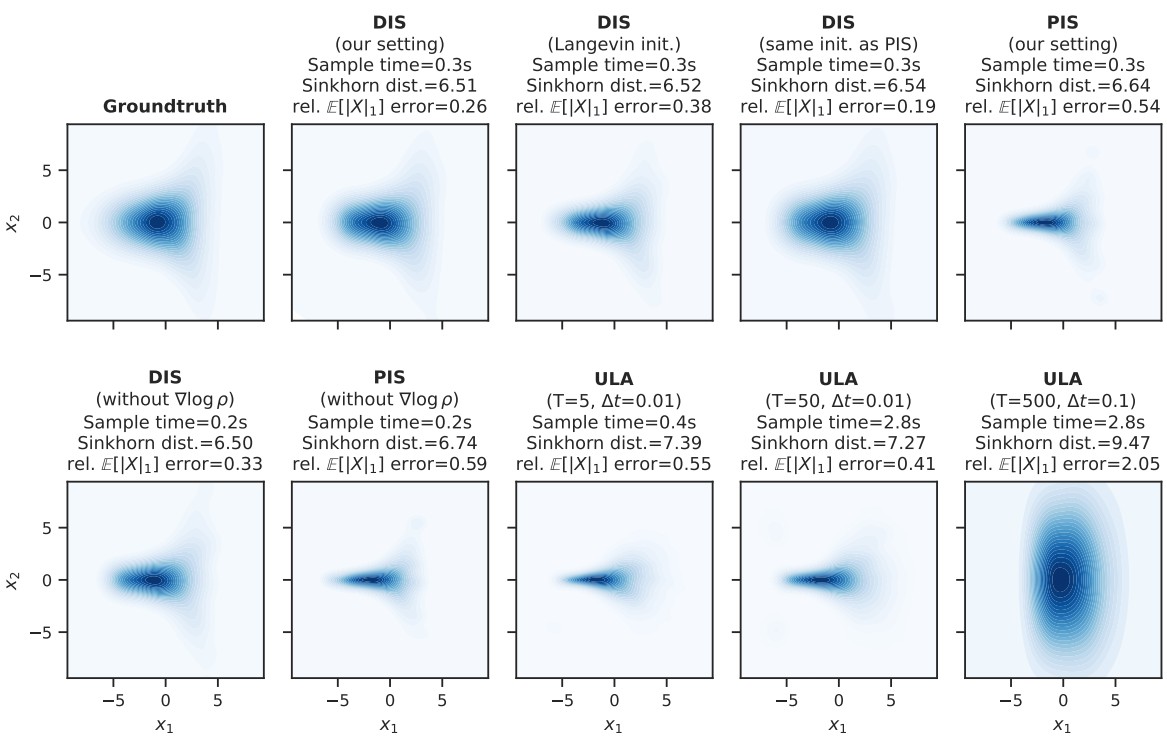

Figure 12: KDE plots visualizing samples from DIS and PIS trained on the Funnel example with different parametrizations, as described in Section A.13. For comparison, we also present results of ULA with two different settings and the runtimes to sample 6000 points on a single GPU.

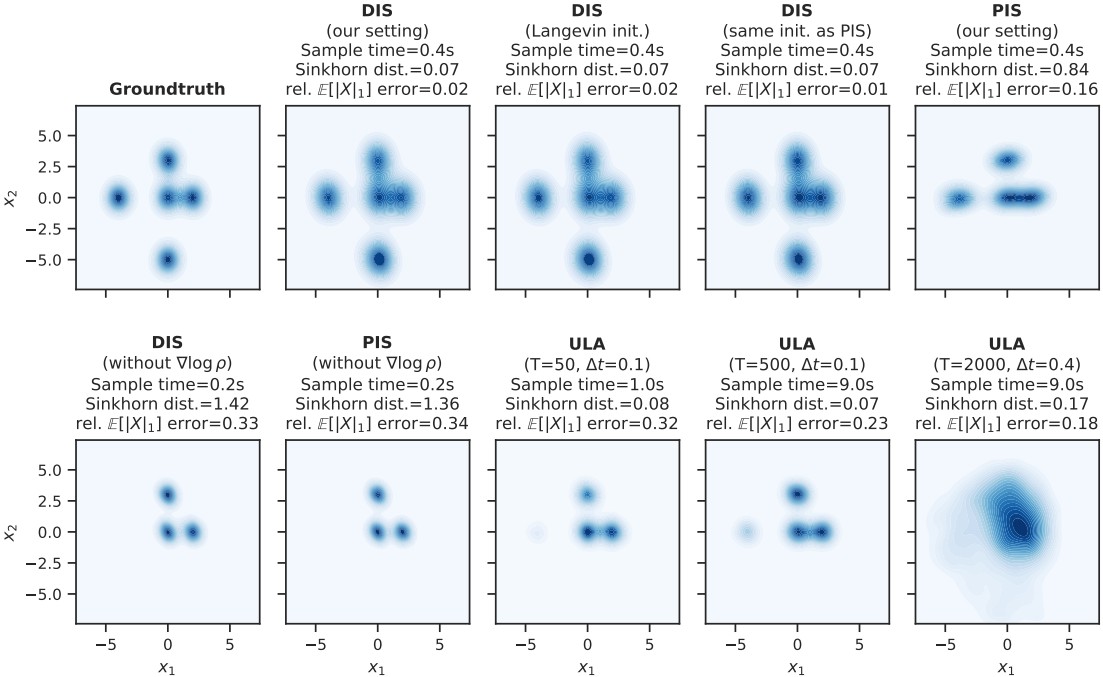

Figure 13: The same experiment as in Figure 12 for a GMM example with means at $(0,0)$, $(0,2)$, $(3,0)$, $(-4,0)$, $(0,-5)$ and covariance matrix $\frac{1}{5}\mathrm{I}$.

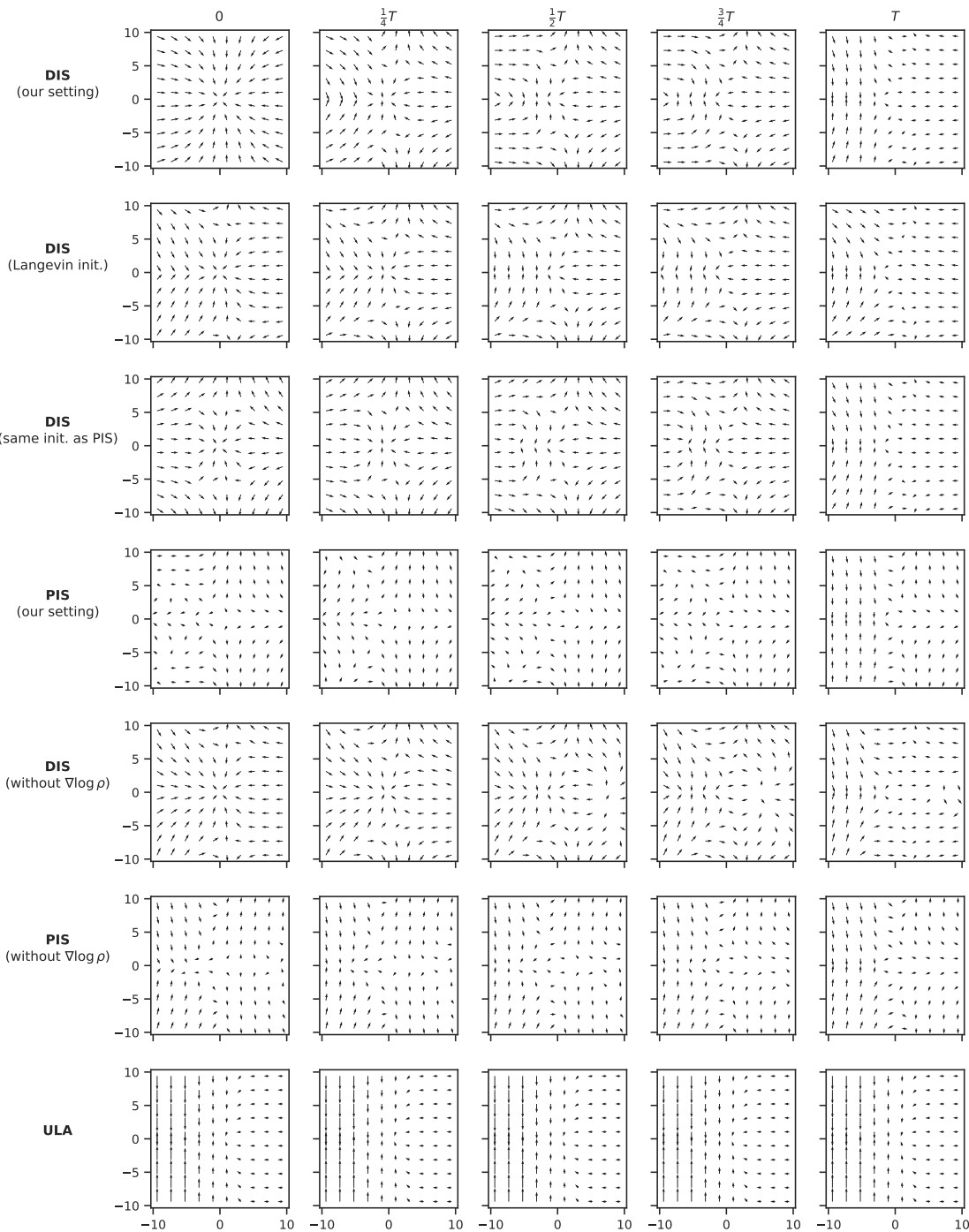

Figure 14: The SDE drifts of the different methods in Figure 12 for $x_1, x_2 \in [-10, 10]$ and $x_3, \ldots, x_{10} = 0$.

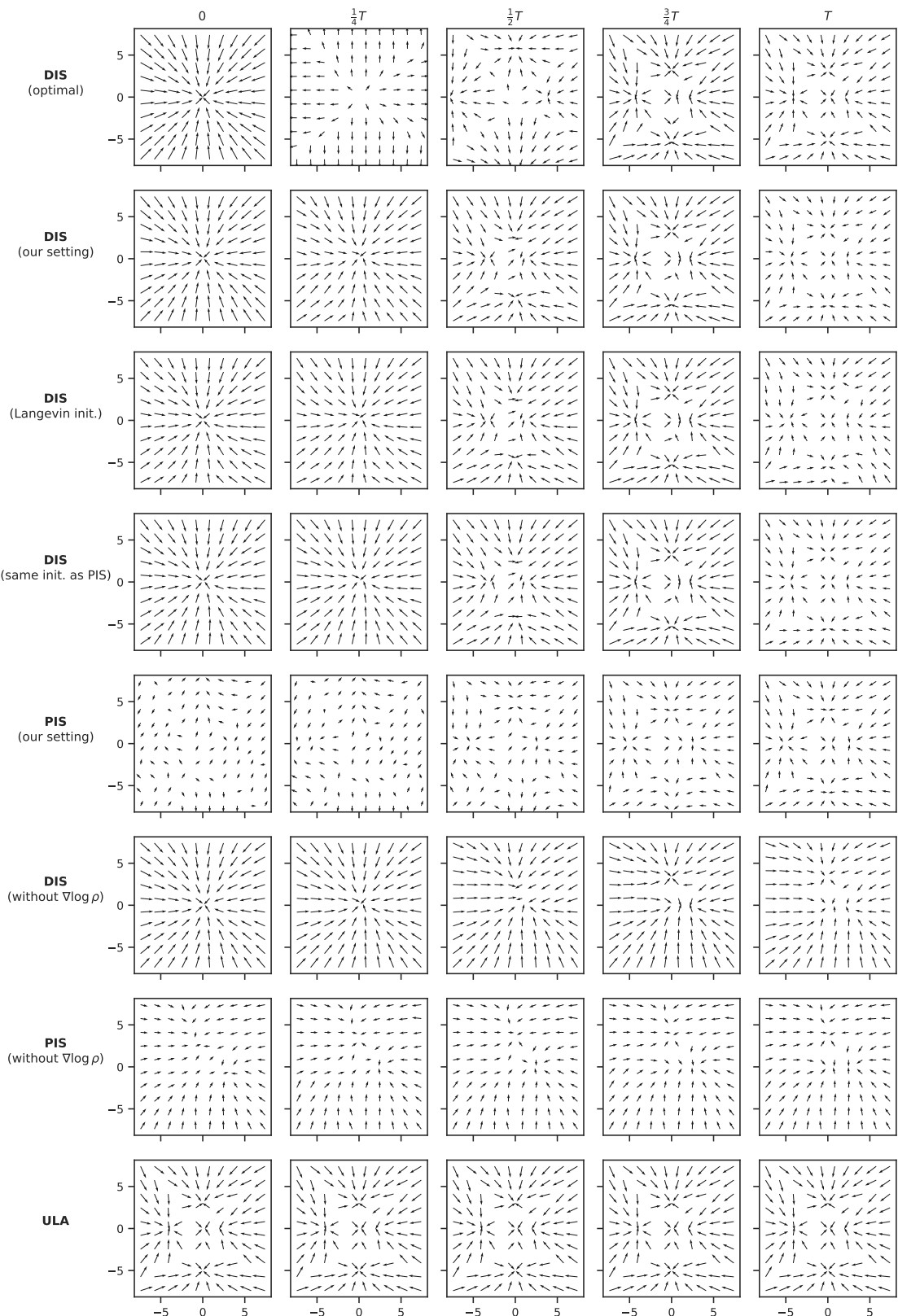

Figure 15: The same visualization as in Figure 14 for the GMM example in Figure 13 and $x_1, x_2 \in [-8, 8]$. We compute the optimal drift for DIS similar to Section A.10.

