# OpenReview forum: "An optimal control perspective on diffusion-based generative modeling"
_TMLR — Accepted by TMLR_

### Review · Reviewer_QKvk · 2023-11-24

**Summary Of Contributions:**

In this paper, the authors establish a connection between generative models built upon general Stochastic Differential Equations (SDEs) and Stochastic Optimal Control (SOC) theory. Given a target distribution $\pi\in \mathcal{P}(\mathbb{R}^d)$ and a horizon time $T>0$, the underlying approach consists in using SOC theory to find a drift $\mu:[0,T]\times\mathbb{R}^d \to \mathbb{R}^d $, a volatility $\sigma:[0,T] \to \mathbb{R}_+$ and a prior distribution $\pi^0\in \mathcal{P}(\mathbb{R}^d)$ such that the SDE given by $dX_t=\mu(T-t, X_t)dt+ \sigma(T-t)dB_t$ with $X_0\sim\pi^0$ verifies $X_T\sim \pi$.

Their method consists in first deriving a Hamilton-Jacobi-Bellman equation for the evolution of the log-densities of the process $(X_{T-t})$. This leads to the formulation of a SOC problem over a specific class of control functions (roughly, additive controls),
which amounts to maximising an evidence lower bound (ELBO) to $\log \pi$. Interestingly, this ELBO connects with the framework of score-based diffusion models [1],[4], and its variational gap can be rewritten as a KL divergence between path measures.

This SOC approach leads to a novel diffusion-based method, referred to as time-reversed diffusion sampler (DIS), to sample from un-normalized densities. In this setting, traditional diffusion-based optimisation tools do not work since one does not have access to samples from $\pi$. In practice, theDIS procedure relies on a gradient-based optimization scheme along sample trajectories, similarly to Path integral Sampler (PIS) [5] and Denoising Diffusion Sampler (DDS) [3].

The authors propose numerical experiments where they apply DIS with reference process chosen as the Ornstein Uhlenbeck process (Variance Preserving scheme in [2]). For this setting, their algorithm recovers the concurrent algorithm from [3]. They compare their method with PIS, and provide better numerical results in various settings.

[1] A variational perspective on diffusion-based generative models and score matching. Huang et al., 2021.

[2] Score-based generative modeling through stochastic differential equations. Song et al, 2020.

[3] Denoising diffusion samplers. Vargas et al, 2023.

[4] A connection between score matching and denoising autoencoders. Vincent, 2011.

[5]  Path integral sampler: a stochastic control approach for sampling. Zhang et al.

**Audience:**

Yes

**Claims And Evidence:**

Yes

**Requested Changes:**

I recommend the authors to introduce a numerical comparison with (i) the implementation from
[3], as it may reveal differences in the numerical scheme, and (ii) standard MCMC schemes. Moreover, I
suggest the authors to run additional experiments in a higher dimensional synthetic setting: for example, Gaussian mixtures
with d = 100, 200.

**Strengths And Weaknesses:**

Strengths:

Although the paper is dense and contains a lot of information, I find it well structured and well written. The
reading is not easy, but I acknowledge that the authors put effort to make it comprehensible. In particular, the details
provided in the appendices are useful to understand the paper.

I think that this paper offers an interesting and strongly grounded perspective on diffusion-based generative modeling. In
particular, the connection with score-based diffusion methods is well explained, and the links between DIS and the related
algorithms is well highlighted.

Besides this, the authors provide lots of details on their numerical implementation, which is key for reproducibility, and
present convincing results in practice.

Weaknesses:

About the sampling method for un-normalized densities: although the authors claim that DIS may be
applied to any general forward process (as long as one can sample from the terminal distribution), the authors only consider
the OU-VP scheme in their experiments. In this case, DIS reduces DDS (see Appendix A.10.1). Then, it is not clear how
DIS is a novel contribution in practice.

In the numerical experiments, a comparison with efficient MCMC methods such as Sequential Monte Carlo is missing.
As showed by [3], these methods may give better results than PIS. Moreover, the highest dimension
considered is not very large (d = 50), compared to previous works.

[3] Denoising diffusion samplers. Vargas et al, 2023.

---

> ### Author Response · Authors · 2023-12-28
>
> Dear Reviewer QKvk,
>
> We thank you for your constructive and helpful review. We are glad about your positive feedback on our theoretical contribution, the numerical experiments, as well as our presentation.
>
> **For the OU-VP scheme, DIS reduces to DDS.**
>
> Recalling that DDS is a special case of DIS, it is indeed true that in these settings DIS is mathematically equivalent to DDS. However, note that the implementations are still different and DIS can yield better performance, see the table below.
> For instance, DIS uses different running and terminal costs compared to DDS.
>
> Furthermore, we think that our derivation is slightly easier as it follows directly from the optimal control perspective and the time-continuous ELBO of score-based generative modeling, whereas DDS relies on introducing a reference process.
>
> **Futher numerical experiments.**
>
> Based on your suggestion, we present experiments for Gaussian mixture models in $d=50,100,200$ in the following:
>
> | Method | GMM-50d | GMM-100d | GMM-200d |
> | --- | --- | --- | --- |
> | PIS | 2.00 | 1.98 | 2.17 |
> | DIS (ours) | **0.20** | **0.32** | **0.91** |
>
> This compares the error in estimating the log-normalizing constant (with importance sampling) using $200$ integrator steps when increasing the dimension of our GMM example (by placing the modes on the hyperplane spanned by the first two dimensions). We clearly observe that the improvements of DIS over PIS also translate to higher-dimensional settings.
>
> Moreover, we emphasize that our proposed DIS method (and its variant DIS-LV using the log-variance loss as described in Appendix A.13) can also outperform other state-of-the-art baselines when computing the log-normalizing constant:
>
> | Method | GMM | Funnel |
> | --- | --- | --- |
> | HMC |  1.876 | 0.835 |
> | NUTS |  1.871 |  0.835 |
> | SMC-1k | 0.362 | 0.338 |
> | SMC-2k | 0.398 | 0.338  |
> | SMC-3k |  0.137 | 0.323 |
> | SMC-5k | 0.129 | 0.298  |
> | SMC-10k | 0.171 | 0.239  |
> | VI-NF  | 1.632 | 0.236 |
> | AFT-1k | 0.509 | 0.249 |
> | AFT-2k  | 0.371 |  0.249 |
> | AFT-3k | 0.316 | 0.281 |
> | AFT-5k  | 0.194 | 0.253 |
> | AFT-10k  | 0.030 | 0.194 |
> | CRAFT | 0.046 | 0.097 |
> | DDS  | 0.223 | 0.186 |
> | PIS | 0.674 | 0.132 |
> | DIS (ours) | 0.021 | 0.032 |
> | DIS-LV (ours) | **0.017** | **0.014**|
>
> For the Funnel and GMM distributions,
> we can compare against the baselines from Zhang \& Chen (2022), i.e., Hamiltonian Monte Carlo (HMC),
> U-Turn Sampler (NUTS),
> Variational inference with normalizing Flows (VI-NF),
> Sequential Monte Carlo (SMC-$n$),
> and Annealed Flow Transport Monte Carlo (AFT-$n$) using $n$ steps. The funnel distribution is also considered by Vargas et al. (2023) and we present their best results for the denoising diffusion sampler (DDS). We further run DDS on the GMM example and also compare against the Continual Repeated Annealed Flow Transport Monte Carlo (CRAFT). Let us know if you would like to see further comparisons and we will include them in our manuscript.

---

### Review · Reviewer_JtqT · 2023-12-05

**Summary Of Contributions:**

The authors study a connection between stochastic optimal control theory and generative models based on SDEs, such as score-based diffusion models. It is shown that the evidence lower bound (ELBO) for SDE-based models follows from a standard result in control theory, and that the expected variational gap of ELBO can be interpreted in terms of the KL divergence between relevant path measures. The authors employ their results to develop a novel sampling algorithm for applications when the target density itself is known up to normalization, but one does not have access to samples. They demonstrate the utility of their sampling algorithm through several numerical experiments.

**Audience:**

Yes

**Claims And Evidence:**

Yes

**Requested Changes:**

Please make changes by addressing the comments on the weaknesses above and some further questions/comments below.

- I am interested to know if you can quantify the sampling error of the DIS algorithm in terms of the estimation error of the control term (i.e., if an analogue of Theorem 2 in [1] holds for your sampling algorithm). Is this something you have considered?
- It is worth to mention that the DIS and PIS algorithms outperform baseline sampling algorithms after having been trained, but that training the models incurs additional run-time. In addition, you might mention any difficulties you encountered during training (and how you dealt with them) in case others want to use your algorithm.
- There have been several recent works (possibly after your submission) using control theory to study generative models, so it would be good to update the related work section. For instance, in [2], the authors expand on the connection between diffusion models and HJB equations using the framework of mean-field games.
- I think a table of contents would improve the readability of the appendix.

[1] Sitan Chen, Sinho Chewi, Jerry Li, Yuanzhi Li, Adil Salim and Anru R Zhang. Sampling is as easy as learning the score: theory for diffusion models with minimal data assumptions. In International Conference on Learning Representations, 2023.
[2] Benjamin Zhang and Markos Katsoulakis. A mean-field games laboratory for generative models. Arxiv preprint Arxiv:2034.13534

**Strengths And Weaknesses:**

Strengths:
- The connections established in the paper may allow for techniques from control theory to be applied to generative models, and vice-versa.
- The analysis in the paper is quite rigorous and appears correct to the best of my knowledge.
- The paper offers an interesting approach to sampling unnormalized densities, which is a central task in many areas of applied mathematics. Their method seems to perform well on several multi-modal distributions, even relative to state of the art sampling algorithms.

Weaknesses:

- I am skeptical of the contributions in Section 2. For instance, I don’t quite see the significance in the fact that the ELBO can be derived from control theory (Theorem 2.4, Corollary 2.5). In addition, the connection between the objective of SDE-based generative models and KL divergence between path measures has been well-studied through the lens of Girsanov’s theorem, for instance in [1] and [2]. It is therefore unclear why the insight from Proposition 2.6 is novel or interesting in the generative modeling context (aside from the fact that it is later used to develop the sampling algorithm).
- The exposition is confusing/difficult to follow at times. It feels like sections 2 and 3 often jump abruptly between several ideas without properly connecting them.
- The DIS sampling algorithm seems quite similar to the path integral sampler proposed in the work of Zhang and Chen [3]. If I understand both correctly, the main differences are in the initial distributions and running/terminal costs, but the connection between sampling and stochastic control seems to have already been exploited. While I appreciate the authors for thoroughly explaining the relationship between the two methods in Section 3.1, it does take away from the novelty of the algorithm.

[1] Yang Song, Conor Durkan, Iain Murray, Stefano Ermon. Maximum likelihood training of score-based diffusion models. In Advances in Neural Information Processing Systems 34, 2021.
[2] Sitan Chen, Sinho Chewi, Jerry Li, Yuanzhi Li, Adil Salim and Anru R Zhang. Sampling is as easy as learning the score: theory for diffusion models with minimal data assumptions. In International Conference on Learning Representations, 2023.
[3] Qinsheng Zhang and Yongxin Chen. Path integral sampler: a stochastic control approach for sampling. In International Conference on Learning Representations, 2022a.

---

> ### Author Response · Authors · 2023-12-28
>
> Dear Reviewer JtqT,
>
> We thank you very much for your extensive review. Let us address your points in the sequel.
>
> **Relevance of contributions in Section 2**
>
> For the relevance of the control perspective, we refer to the end of our introduction section, which lists the most important points. In the general response we further elaborate on these aspects.
>
> We note that the works [A] and [B] differ from our work regarding the connections between SDE-based generative modeling and KL divergences between path measures. In particular, these works do not explicitly state the target path measure and only find expressions which still include the score function. However, the latter is not known in practice and such expressions can therefore not be used for practical implementations. On the other hand, in our Proposition 2.6, the optimal path measure is stated explicitly and does not include the score function. In consequence, it readily allows for concrete algorithms based on arbitrary divergences as demonstrated in our numerical examples.
>
> **DIS is similar to PIS**
>
> DIS and PIS are indeed similar in the sense that both rely on controlled diffusions and aim to sample from a prescribed target density. However, we highlight that DIS is directly linked to score-based generative modeling, whereas PIS is not. This connection allows to carry over knowledge from score-based generative modeling to sampling -- both in terms of theory and practical implementations. Note that we also discuss numerical advantages of DIS over PIS in Section A.10.2 in the appendix.
>
> **Quantification of sampling error of the DIS algorithm.**
>
> Theorem 2 in [B] indeed holds in our setting -- the data distribution is just replaced by an explicitly given density -- and we added this reference in our manuscript (together with the related references [C,D]).
> Recall that (up to scaling) the score function corresponds to the optimal control. Therefore, the error $\varepsilon_\mathrm{score}$ in [B] immediately translates to the error of the control function $u$ w.r.t. to the optimal control $u^*$. Using the results in [B], one can thus quantify the sample quality assuming that the optimal control is $L$-Lipschitz and the $L^2$-error to the optimal control is bounded. In particular, under mild regularity assumption, a total variation distance to the target distribution can be achieved when using $\mathcal{O}(L^2d/\varepsilon)$ steps. In particular, this holds without any log-concavity assumption on the target, but still matches state-of-the-art complexity bounds for Langevin Monte Carlo algorithms applied to targets satisfying a log-Sobolev inequality.
>
> **Include some recent work.**
>
> Thank you for pointing out the work [E], which was indeed developed after our work. We included it in the updated version of our paper.
>
> **Table of contents.**
>
> We included a table of contents for the appendix in the updated version of the manuscript.
>
> **Highlight aspect of training PIS and DIS more.**
>
> Thank you for this suggestion. We added a "limitations" section, see Section 4.3, where we clarify that -- in contrast to more classical sampling approaches -- our sampling methods rely on sufficient training.
>
> ---
>
> [A] Yang Song, Conor Durkan, Iain Murray, Stefano Ermon. Maximum likelihood training of score-based diffusion models. In Advances in Neural Information Processing Systems 34, 2021.
>
> [B] Sitan Chen, Sinho Chewi, Jerry Li, Yuanzhi Li, Adil Salim and Anru R Zhang. Sampling is as easy as learning the score: theory for diffusion models with minimal data assumptions. In International Conference on Learning Representations, 2023.
>
> [C] Lee, H., Lu, J., & Tan, Y. (2022). Convergence of score-based generative modeling for general data distributions. arXiv preprint arXiv:2209.12381.
>
> [D] De Bortoli, V. (2022). Convergence of denoising diffusion models under the manifold hypothesis. Transactions on Machine Learning Research.
>
> [E] Benjamin Zhang and Markos Katsoulakis. A mean-field games laboratory for generative models. Arxiv preprint Arxiv:2034.13534

---

### Review · Reviewer_A8EE · 2023-12-12

**Summary Of Contributions:**

The main contribution of this paper is a connection between diffusion probabilistic models (DPMs) and stochastic optimal control: the authors prove that score-based generative modelling with stochastic differential equations (SDEs) (Song et al., 2020b) amounts to solving a Hamilton-Jacobi-Bellman (HJB) equation (Lemma 2.3). This enables drawing the connection with stochastic optimal control: the HJB equation governing score-based generative models is linked to a controlled SDE (Theorem 2.4). While this result is shown by applying a classical theorem from control theory (the verification theorem), it offers a new perspective on score-based generative models, which are usually associated with the Fokker-Planck equation.

The authors then leverage this connection to derive a lower bound on the negative log-likelihood of the considered generative model (Corollary 2.5). This bound is interpreted as an evidence lower bound and justifies another perspective of score-based generative modeling as a variational inference problem on the space of continuous trajectories (Proposition 2.6). Based on this result, the authors finally develop a new procedure called "time-reversed diffusion sampler" to sample from distributions with intractable normalizing constant (Section 3) and illustrate its relevance on toy experiments (Section 4).

**Audience:**

Yes

**Claims And Evidence:**

Yes

**Requested Changes:**

Given my points above, I would encourage the authors to clarify the insights offered by their stochastic optimal control approach for diffusion models. In particular,

- There is enough explanations on the mathematical derivations in my opinion: what is missing is intuition. What does this optimal control perspective on diffusion model bring concretely? Why does the proposed sampling procedure outperform PIS? Is the perspective useful for classical experiments for diffusion models, e.g., high-dimensional image generation?

- The second equality of eq. (5) is crucial for the rest of the paper, but is not proved anywhere. I confirm it is correct (I checked it using identities in Appendix A.2) but I strongly encourage the authors to give the detailed derivations in their paper.

- The authors provide some background on diffusion models in the main text, but not on stochastic optimal control (this is provided in Appendix A.6). Moving some of these elements from the appendix to the main text might help the reader understand why it is useful to use controlled SDEs for diffusion models.

**Strengths And Weaknesses:**

**Strengths**

- This paper bridges the gap between different fields (generative modelling, approximate Bayesian inference, control theory) to offer a perspective on diffusion models which, to the best of my knowledge, is novel.

- The main contribution is rigorously justified (it is obtained by manipulating PDEs and applying results from control theory) and can be applied to develop practical, theoretically-grounded methodologies, as illustrated with the sampling procedure in Section 3. In that sense, I believe this is relevant for the machine learning community and can foster future interesting work.

**Weaknesses**

- Although the paper is well written overall, it suffers from a lack of clarity in my opinion. The paper is quite dense and high-level/more intuitive explanations on the contributions are missing. For instance, the main results are provided in Sections 2.2 and 2.3, and the authors provide detailed explanations on their mathematical technicality, which I appreciate (e.g., see paragraph below Corollary 2.5) -- but I am missing the big picture, which is: why does it help to view a diffusion model as a controlled SDE? What are the "valuable new insights" (Conclusion) on diffusion models provided by control theory? The fact that the authors then parameterize the control $u$ as a neural network, which are black-box algorithms, does not help clarify.

- The experiments are a bit disappointing: given that this paper strongly relies on score-based generative models, I would have expected experiments similar to the ones in (Song et al., 2020b) to be conducted. It would be interesting to understand why the authors have chosen to not explore the consequences of their optimal control perspective on classical diffusion experiments, such as image generation. Besides, regarding the experiments in Section 4, the results are indeed encouraging but no insight is given on why DIS outperforms PIS.

---

> ### Author Response · Authors · 2023-12-28
>
> Dear Reviewer A8EE,
>
> We thank you for your review. Your summary of our work appears very adequate and we appreciate that you value our contribution as novel.
>
> **Relevance of control perspective.**
>
> For the relevance of the control perspective we refer to the end of our introduction section, which lists the most important points. We further elaborate on these aspects in our general response. Feel free to let us know if you are missing certain aspects and we will change the manuscript accordingly.
>
>
> **Derivation of Kolmogorov backward equation.**
>
> Thank you for this suggestion, we have added the computation in Appendix A.5. We note that the Kolmogorov backward equation (i.e. the second equation in Eq. (5)) readily follows from the Fokker-Planck equation (which is the first equation in Eq. (5)) by applying the divergence identities outlined in Appendix A.2, as explained after Eq. (5).
>
> **Introduction to stochastic optimal control.**
>
> To improve readability, we added a reference to Appendix A.6 (where we provide an introduction to stochastic optimal control theory) in Section 2.2. Note that we also provide intuition on the optimal control perspective in the introduction, e.g., highlight its relevance for generative modeling. However, for the sake of readability and given the limited space in the main text, we prefer to keep the introductory material in the appendix.
>
> **Choice of experiments.**
>
> As correctly observed by you, we have indeed focused on the setting where only an unnormalized density is specified but no samples from the target distribution are available (as, e.g., in image generation). We think that this setting can be even more challenging, e.g., due to exploration-exploitation trade-offs and mode collapse. We agree that the other case has been studied extensively in the past years, e.g., by Song et al. (2020b). The novel algorithms that we have so far deduced from the control and path space perspectives, however, require the knowledge of the target density in closed form (at least up to the normalization constant). Only in this setting, one can directly minimize divergences between path space measures, such as the KL or log-variance divergence. If the density is not available one must resort to the ELBO, as commented on in Section 2.4. We have added Section 4.3 to address some of the limitations of our work more explicitly.

---

### Review · Reviewer_ziUJ · 2023-12-13

**Summary Of Contributions:**

The authors present an optimal control perspective on diffusion based generative modelling. They formulate the minimisation of the evidence lower bound within a stochastic control formalism. By appealing to the Girsanov theorem for the change of measure they compute  the optimal change of measure through the Doob-h transform. This allows to express the evidence lower bound (ELBO) of the generative model in terms of the control costs of the stochastic control problem.
In the second part of the paper they proceed with providing an estimator (time-reversed diffusion sampler - DIS) for sampling from unnormalised probability densities. They conduct numerical comparisons between this proposed approach and the recent path integral sampler developed by Zhang & Chen in 2022.

The paper provides an interesting perspective between diffusion-based generative modeling and stochastic control. While it doesn't unveil anything entirely new, it sheds light on a connection that, as far as I know, hadn't been explicitly articulated before, except for a mention in [7].

**Audience:**

Yes

**Claims And Evidence:**

Yes

**Requested Changes:**

- Please include in the relative work some overview on optimal stochastic control and Kulback-Leibler control for stochastic systems, that uses the KL divergence from a reference path measure to compute the controls, similar to the presented approach, and also the other points mentioned in the weaknesses.

- In Figure 4, both PIS and DIS seem to perform particularly bad in approximationg the log evidence for the double well system. Can you provide some intuition behind this.
Also how would the method compare with other methods for computing the evidence, like in [10]?

**Strengths And Weaknesses:**

**Strengths:**

- I find interesting the perspective of using different (than the KL) divergences between the path measures of the reference and the constrained diffusion process for crafting alternative algorithms for training. Their experiments suggest that using the log-variance divergence leads to more stable estimates of the log evidence with less valiance.


- I find the connections the authors provide with denoising score matching and half-bridge training for solving the Schroedinger bridge problem valuable. However, these parts are probably more fitting for the appendix (especially the former one). In this version they distract the reader from the main message/contribution of the paper.


**Weaknesses:**

- While Pavon's work in [7]  already makes the association between diffusion based generative modelling and stochastic control, I acknowledge that the authors provide here a more well-presented and comprehensive version of the association. However they should should acknowledge this existing work of Pavon. They could emphasize their contributions by highlighting their additional contributions.

- On the related work the authors do not make any mention to the extensive optimal stochastic control literature that is a crucial part of their framework, i.e., [1] - [6]. For example [4] uses a forward and time-reversed diffusion process to compute optimal controls for SDEs starting from Dirac delta functions. Their derivation follows similar arguments with minimising the KL divergence between a reference process and the controlled process.

- The association between minimisation of the lower bound for diffusion processes and optimal control has been already proposed for inference of diffusion processes with variational inference in [8] and [9], where in [9] the authors consider a forward and time-reversed process akin to diffusion-based generative modeling to compute the optimal controls that minimise the ELBO.

- I find the writing and presentation style of the numerical experiments a bit confusing. I recommend the authors to rewrite this section to provide initially with a summary and a motivation what they attempt to demonstrate with each experiment and then presentation and walkthrough the results. In the current version, the section 4.1 Examples fits better to an Appendix, while in the results section of the main text I would mention the name of the models employed, why these were selected and then list the results for each.

- The authors do not mention relevant literature on sampling unnormalised densities, i.e. energy based models.


[1] Kappen, Hilbert Johan, and Hans Christian Ruiz. "Adaptive importance sampling for control and inference." Journal of Statistical Physics 162 (2016): 1244-1266.

[2] Thijssen, Sep, and H. J. Kappen. "Path integral control and state-dependent feedback." Physical Review E 91.3 (2015): 032104.

[3] Kappen, Hilbert J., Vicenç Gómez, and Manfred Opper. "Optimal control as a graphical model inference problem." Machine learning 87 (2012): 159-182.

[4] Maoutsa, Dimitra, and Manfred Opper. "Deterministic particle flows for constraining stochastic nonlinear systems." Physical Review Research 4.4 (2022): 043035.

[5] Zhang, Wei, et al. "Applications of the cross-entropy method to importance sampling and optimal control of diffusions." SIAM Journal on Scientific Computing 36.6 (2014): A2654-A2672.

[6] Hartmann, Carsten, and Christof Schütte. "Efficient rare event simulation by optimal nonequilibrium forcing." Journal of Statistical Mechanics: Theory and Experiment 2012.11 (2012): P11004.

[7] Pavon, Michele. "On Local Entropy, Stochastic Control, and Deep Neural Networks." IEEE Control Systems Letters 7 (2022): 437-441.

[8] Opper, Manfred. "Variational inference for stochastic differential equations." Annalen der Physik 531.3 (2019): 1800233.

[9] Maoutsa, Dimitra. "Geometric constraints improve inference of sparsely observed stochastic dynamics." International Conference on Learning Representations (ICLR) 2023 - Workshop on Physics for Machine Learning (Physics4ML) (2023).

[10] Zaki, Nikolai, Théo Galy-Fajou, and Manfred Opper. "Evidence Estimation by Kullback-Leibler Integration for Flow-Based Methods." Third Symposium on Advances in Approximate Bayesian Inference. 2020.

---

> ### Author Response · Authors · 2023-12-28
>
> Dear Reviewer ziUJ,
>
> We thank you for your helpful review. We are glad that you value our novel connection between diffusion-based generative modeling and stochastic optimal control, as well as the fact that our framework allows to employ arbitrary divergences, e.g., the log-variance divergence. We respond to your questions and concerns in the sequel.
>
> **Pavon's work in [7] already makes the connection between diffusion based generative modelling and stochastic control.**
>
> First, note that we have already cited Pavon's work [7] in our manuscript. We have additionally referred to [A] and [B] of Pavon, which seem even more related to our setting. As far as we can see, [7] "only" relates stochastic optimal control to the *Schrödinger half-bridge problem* --- we emphasize this in the revised version of our manuscript. Note, however, that the connection of half-bridges to optimal control goes back to the work [C] and has also already been observed in [D-F] (as cited in our paper).
>
> **Refer to more optimal control literature.**
>
> Thank you for providing additional references. The work [4] indeed also uses time-reversed SDEs and appears to be similar to our approach. We now cite it in our revised version and we also refer to the work [9]. Moreover, we have added two additional references that relate to the connection of path space measures and optimal control theory.
>
> Finally, we note that stochastic optimal control theory dates back to the 1970s. We already cited multiple relevant text books, such as [G], originally published in 1975. Additionally, we provide a self-contained primer on stochastic control theory in appendix A.6.
>
> **Mention relevant literature on sampling unnormalized densities, i.e. energy based models.**
>
> Note that we have referred to multiple alternative sampling strategies in Section 1.1, e.g., MCMC methods or normalizing flows. If you have any specific method in mind, we would be happy include it in our related work section. Note that we also provide a empirical comparison of other methods in the response to reviewer QKvk.
>
> **Bad performance of PIS and DIS in Figure 4 for the double well example.**
>
> We note that this is a particularly challenging example featuring $32$ very separated ($\delta=4$) modes in $d=20$ dimensions.
>
>
> ---
>
> [A] Michele Pavon. Stochastic control and nonequilibrium thermodynamical systems. Applied Mathematics and
> Optimization, 19(1):187–202, 1989.
>
> [B] Paolo Dai Pra and Michele Pavon. On the markov processes of Schrödinger, the Feynman-Kac formula and
> stochastic control. In Realization and Modelling in System Theory, pp. 497–504. Springer, 1990.
>
> [C] Paolo Dai Pra. A stochastic control approach to reciprocal diffusion processes. Applied mathematics and
> Optimization, 23(1):313–329, 1991.
>
> [D] Belinda Tzen and Maxim Raginsky. Theoretical guarantees for sampling and inference in generative models
> with latent diffusions. In Conference on Learning Theory, pp. 3084–3114. PMLR, 2019.
>
> [E] Lorenz Richter. Solving high-dimensional PDEs, approximation of path space measures and importance
> sampling of diffusions. PhD thesis, BTU Cottbus-Senftenberg, 2021.
>
> [F] Qinsheng Zhang and Yongxin Chen. Path integral sampler: a stochastic control approach for sampling. In
> International Conference on Learning Representations, 2022a.
>
> [G] Wendell H Fleming and Raymond W Rishel. Deterministic and stochastic optimal control, volume 1. Springer
> Science \& Business Media, 2012.

---

### Author Response · Authors · 2023-12-28
**General response**

We thank all reviewers for their helpful reviews and constructive feedback that helped us to further improve our manuscript. We updated the PDF accordingly and tried to address all remaining questions in our individual responses to the reviewers.

In this general response, we want to further highlight the significance of our findings and outline how they lead to new methods and insights in diffusion-based generative modeling and sampling. In particular, we want to mention the following perspectives:

- *PDE perspective*: The HJB PDE that we have identified as the governing PDE for the score function (Lemma 2.3) might be useful for further theoretic analyses (e.g., the behavior of the score function for different noise schedules), but also for practical algorithms for the numerical approximation of the score, based on neural PDE solvers, such as PINNs or BSDEs, see A.11. We note that PINNs do not rely on space and time discretizations and we demonstrated in Figure 7 that they constitute a competitive method.

- *Control perspective*: One can now readily apply numerical methods from control theory to approximate the score function, e.g. policy iteration, which has more recently been combined with deep learning [1] and methods based on tensor trains (in contrast to neural networks), see e.g. [2].
Note that one usually controls a linear (Ornstein-Uhlenbeck type) process. If one approximates the target density with Gaussians first (which corresponds to quadratic costs in the control problem), one could "pretrain" the score model by solving a linear-quadratic-control problem first, for which very efficient numerical methods have been developed extensively in the last decades.

- *Sampling*: Our control perspective directly yields a novel, diffusion-based method to sample from unnormalized densities. Our DIS method can already outperform PIS, the previous state-of-the-art in diffusion-based sampling. Moreover, the connection to diffusion models allows to transfer successful noise schedules, integrators, and techniques (such as the probability flow ODE) from generative modeling to sampling. Furthermore, we show that DIS provides a generalization of DDS (A.10.1).

- *Path space perspective*: This perspective on diffusion models provides an intuitive representation of the objective and the variational gap. More importantly, it allows us to consider arbitrary divergences on path space, such as the log-variance divergence, leading to potentially advanced loss functions and algorithms. In the appendix and in the answer to reviewer QKvk, we show that this has the promise of providing further improvements for DIS.

In summary, our view allows to transfer results between generative modeling, sampling, and optimal control, which brings both new theoretical results (underlying HJB PDE, path measure representation, control objective) as well as new practical algorithms (DIS, neural PDE solver for sampling, usage of other divergences).

---

[1] Mo Zhou, Jiequn Han, and Jianfeng Lu. Actor-critic method for high dimensional static hamilton–jacobi–
bellman partial differential equations based on neural networks. SIAM Journal on Scientific Computing,
43(6):A4043–A4066, 2021

[2] Mathias Oster, Leon Sallandt, and Reinhold Schneider. Approximating optimal feedback controllers of finite
horizon control problems using hierarchical tensor formats. SIAM Journal on Scientific Computing, 44(3):
B746–B770, 2022

---

### Author Response · Authors · 2024-02-25
**Camera Ready Revision**

Dear Editor and Reviewers,

We thank you for your helpful reviews and comments which significantly improved our manuscript. As suggested, we added further discussion and empirical comparisons between DDS and DIS in Appendix A.10.1.

---

### Decision · Action_Editor_RYjp · 2024-01-24

**Recommendation:** Accept with minor revision

**Comment:**

All the reviewers agree that the optimal control point of view on diffusion models and related algorithms is valuable for the community. The interaction between diffusion models, sampling and the stochastic optimal control frameworks are very promising. The authors have addressed the main remarks of the reviewers in a convincing manner. I therefore recommend the manuscript for acceptance.  However, I would have liked to see a more detailed comparison between DDS [1] and DIS. In particular, it would be good to highlight where DIS shines and DDS fails.

[1] Denoising diffusion samplers. Vargas et al, 2023.

**Audience:**

Yes.

**Claims And Evidence:**

Yes.